# A back-door insight into the modulation of Src kinase activity by the polyamine spermidine

**Sofia Rossini[1], Marco Gargaro[1], Giulia Scalisi[1], Elisa Bianconi[2], Sara Ambrosino[1], Eleonora Panfili[1], Claudia Volpi[1], Ciriana Orabona[1], Antonio Macchiarulo[2], Francesca Fallarino[1], Giada Mondanelli[1]\***

[1]Department of Medicine and Surgery, University of Perugia, Perugia, Italy; [2]Department of Pharmaceutical Sciences, University of Perugia, Perugia, Italy

**Abstract** Src is a protein tyrosine kinase commonly activated downstream of transmembrane receptors and plays key roles in cell growth, migration, and survival signaling pathways. In conventional dendritic cells (cDCs), Src is involved in the activation of the non-enzymatic functions of indoleamine 2,3-dioxygenase 1 (IDO1), an immunoregulatory molecule endowed with both catalytic activity and signal transducing properties. Prompted by the discovery that the metabolite spermidine confers a tolerogenic phenotype on cDCs that is dependent on both the expression of IDO1 and the activity of Src kinase, we here investigated the spermidine mode of action. We found that spermidine directly binds Src in a previously unknown allosteric site located on the backside of the SH2 domain and thus acts as a positive allosteric modulator of the enzyme. Besides confirming that Src phosphorylates IDO1, here we showed that spermidine promotes the protein–protein interaction of Src with IDO1. Overall, this study may pave the way toward the design of allosteric modulators able to switch on/off the Src-mediated pathways, including those involving the immunoregulatory protein IDO1.

**\*For correspondence:** giada.mondanelli@unipg.it

**Competing interest:** The authors declare that no competing interests exist.

## Editor's evaluation

This is an important study describing the mechanism of Spermidine modulation of Src kinase, identifying the interacting amino acids and the effect on indoleamine 2,3-dioxygenase 1 (IDO1) activation based on solid evidence. Considering the important role of IDO1 in the immune response this study could provide important information for the design of allosteric modulators capable of turning SRC on/off.

## Introduction

Besides being intermediates in metabolic reactions, metabolites can serve as intra- and intercellular signals (*Piazza et al., 2018*). Indeed, by interacting with specific molecular partners, soluble mediators can trigger a series of molecular events critical for cell fitness and adaptation. Metabolites binding to either the active site or the allosteric pocket – that is, that different from the catalytic site – of enzymes are among the best-characterized interactions that modulate protein activity as well as the assembly and function of multiprotein complexes (*Changeux and Christopoulos, 2016*; *Mondanelli et al., 2020b*; *Feng et al., 2014*).

The naturally occurring polyamines (i.e., putrescine, spermidine, and spermine) are organic cations derived from the decarboxylation of L-ornithine, which is generated by the arginase 1 from L-arginine (*Pegg, 2016*; *Igarashi and Kashiwagi, 2000*). The conversion of L-ornithine into putrescine is

catalyzed by the rate-limiting enzyme ornithine decarboxylase 1, followed by two specific synthases that sequentially give rise to spermidine and spermine (*Pendeville et al., 2001*). These metabolites are protonated at physiological pH levels, allowing them to interact with negatively charged macro-molecules, including nucleic acids, proteins, and phospholipids. Given their structure, polyamines indeed modulate several cellular processes, ranging from cell growth and proliferation to immune system function (*Proietti et al., 2020*; *Holbert et al., 2022*). As a matter of the fact, alteration of polyamines intracellular content is associated with the occurrence of several tumors, including prostate, breast, and colon cancers, for which polyamines are considered as biomarkers (*Gerner et al., 2018*; *Geck et al., 2020*; *Nakkina et al., 2021*). Among polyamines, spermidine has recently gained much more attention as player of immune regulation and in age-related disorders, such as cardiac hypertrophy and memory impairment (*Madeo et al., 2018*; *Li et al., 2020*; *Eisenberg et al., 2016*; *Yang et al., 2016*; *Ni and Liu, 2021*; *Liu et al., 2020*). Spermidine exerts a protective role in mouse experimental models of autoimmune diseases, such as multiple sclerosis and psoriasis, by activating the Forkhead box protein O3 (FOXO3) pathway and thus suppressing the production of inflammatory cytokines tumor necrosis factor (TNF)-α and interleukin (IL)-6 (*Li et al., 2020*). More-over, spermidine is able to reprogram mouse conventional dendritic cells (cDCs) toward an immu-noregulatory phenotype via Src kinase-dependent phosphorylation of indoleamine 2,3-dioxygenase 1 (IDO1) (*Mondanelli et al., 2017*). However, the exact mechanism of Src activation by spermidine remains to be elucidated.

The non-receptor tyrosine kinase Src is the representative of a family of structure-related kinases initially discovered as a proto-oncogene regulating critical cellular functions (*Oppermann et al., 1979*). Src activation mainly occurs downstream of multiple transmembrane receptors, including epidermal growth factor receptor (EGF-R), fibroblast growth factor receptor (FGF-R), and insulin-like growth factor-1 receptor (IGF-1R). Indeed, a dysregulated Src activity has been associated with tumor growth and metastasis, inflammation-mediated carcinogenesis, and therapeutic resistance to tradi-tional antineoplastic drugs (*Caner et al., 2021*; *Ahn et al., 2018*; *Cardin et al., 2018*; *Dosch et al., 2020*; *Roskoski, 2015*; *Shao et al., 2022*). The induction of Src kinase activity can also occur following Aryl hydrocarbon Receptor (AhR) activation, whose conformational changes favor the Src-AhR disjunc-tion, allowing the former to phosphorylate its downstream partner IDO1 and thereby promote the generation of an immunoregulatory milieu (*Manni et al., 2020*).

In addition to the kinase domain, Src possesses an N-terminal Src homology-4 (SH4) domain, a unique domain, an SH3 domain, an SH2 domain, an SH2-kinase linker domain, and a C-terminal auto-regulatory motif (*Roskoski, 2015*). The SH2 and SH3 modules serve in protein–protein interactions that are essential for the regulation of kinase activity and signaling function. Specifically, the SH2 domain contains two distinct binding pockets. The first one has a conserved arginine residue that binds a phosphotyrosine (pY) residue presented by the protein substrate, whereas the second pocket binds the residue that is three positions C-terminal of pY (pY+3), contributing to the specificity in ligand–protein recognition.

The autoregulatory function of the kinase occurs through intramolecular interactions that stabilize the catalytically inactive conformation of Src, in which the SH2 domain binds to a pY located at position +530 of the human sequence. Accordingly, binding of ligand proteins to the SH2 domain displaces intramolecular contacts and promotes the catalytic activation of the Src kinase, which is characterized by the phosphorylation of a tyrosine residue in the activation loop (Y419 of the human sequence). Given the crucial role of the non-catalytic domains in modulating Src kinases activity, efforts have been made to develop drug-like modulators of the SH2 and SH3 domains. Small peptidomimetics destabilize the closed conformation and thus promote the kinase activation through the binding of SH3 and/or SH2 domains (*Moroco et al., 2015*; *Moroco et al., 2014*). Alternatively, modulators of Src kinases able to reinforce the intramolecular interactions have proven to allosterically inhibit the enzyme activity (*Dorman et al., 2019*).

Prompted by the finding that spermidine triggers the immunosuppressive IDO1 signaling in cDCs (*Mondanelli et al., 2017*), here we investigated the molecular relationship between that polyamine, Src kinase and IDO1. We found that spermidine (1) activates Src kinase with an allosteric mechanism; (2) binds directly Src kinase at a previously unknown allosteric site; (3) favors the association of IDO1 and Src kinase.

## Results

### Spermidine causes allosteric activation of the kinase activity of Src

The activation of Src kinase mainly occurs downstream of multiple transmembrane and intracellular receptors (such as AhR) as well as protein tyrosine phosphatases (*Manni et al., 2020*; *Mondanelli et al., 2020a*; *Arias-Romero et al., 2009*). In cDCs, it has been demonstrated that a small molecule, namely spermidine, activates Src with a still undefined mechanism (*Mondanelli et al., 2017*). To figure out whether a direct activation would occur, we assayed spermidine against purified recombinant human Src (rhSrc) protein. After 30 min of incubation, a luminescent assay was used to measure the ADP released by the kinase. Results showed that spermidine activated rhSrc with a half-maximal effective concentration (EC50) of 106.4 ± 13.4 nM (*Figure 1A*). To confirm the modulation of the kinase also in living cells, we resorted to immunoblot analysis of phosphorylated murine Src at the tyrosine Y418 as sign of kinase activation. SYF cells that is fibroblast null for Src family kinases, Src, Yes, and Fyn (*Klinghoffer et al., 1999*) were stably transfected with vector encoding for murine Src kinase and then treated with increasing concentration of spermidine. Results showed that the metabolite promoted Src phosphorylation with an EC50 of 6.4 ± 0.6 µM (*Figure 1B, C*). In addition, by measuring the kinase activity in cells endogenously expressing Src (i.e., the murine colon cancer cell line MC38), we confirmed the ability of the polyamine to activate Src (*Figure 1D*).

To get insights into the mechanism of action of spermidine, we measured the intrinsic activity of the polyamine in the absence of either ATP or the synthetic peptide. Results showed that spermidine did not activate Src in the absence of either ATP or peptide (*Figure 1E*), while it promoted the production of ADP when the substrate is also present, ruling out any competition for the same site. As this profile was compatible with an allosteric modulation, we incubated rhSrc with different concentration of ATP or peptide. In the presence of fixed amount of spermidine and increasing concentration of the peptide, the maximum rate of Src kinase activity (Vmax) and the affinity (Km) for the substrate increase (*Figure 1F*). On the contrary, in the presence of different concentration of ATP, spermidine did not affect neither the efficacy nor the affinity of Src kinase (*Figure 1—figure supplement 1*). Such a kinetic profile is consistent with a non-ATP competition, suggesting that spermidine allosterically activates the kinase activity of Src. On measuring the spermidine effect on constitutive active Src, we resorted to the SYF cells model expressing the Src carrying a tyrosine to phenylalanine mutation at position 529 of the murine sequence (here, Src Y529F). Results demonstrated that the polyamine is not able to activate Src Y529F as measured by its phosphorylation (*Figure 1—figure supplement 1*), suggesting that spermidine per se cannot activate or stabilize the constitutive active form of Src, instead it might promote the conformational changes of the kinase and thus its activation.

### Spermidine binds to a negatively charged pocket in SH2 domain of Src kinase

In the inactive state, Src assumes a closed conformation with the SH3 domain bound to the SH2-kinase linker and the SH2 domain bound to the tyrosine phosphorylated tail (*Figure 2A*). Using the experimental available structure of single SH2 domain (PDB ID: 2JYQ; viral isoform), we characterized key structural and electrostatic elements involved in ligand/protein recognition using electrostatic potential calculations (*Figure 2B*). A positive electrostatic potential was observed in the region of the pY-binding site (R178 and H204 residues according to sequence numbering of human isoform, *Figure 2B*), whereas a stretch of surface endowed with a strong negative electrostatic potential was observed on the backside of the pY-binding site as delimited by glutamate residues E150 and E169 (*Figure 2B*), suggesting the existence of a putative allosteric site. Of note, by the alignment of amino acid sequences, we identified that such residues were conserved in Src protein of human, murine and viral isoforms (*Figure 2—figure supplement 1*), further supporting potential functional role for this allosteric site.

A docking study was carried out to investigate the binding mode of spermidine into the allosteric site of Src SH2 domain. As a result, n.18 solutions were obtained showing a conserved binding mode located in a shallow cavity close to E150 and shaped by A148, F153, and T250 (sequence numbering of human isoform, *Figure 2C*). According to the top scored solution (*Supplementary file 1*; *Figure 2D*), the first primary amine group interacts by an electrostatic enforced hydrogen bond with E150, the secondary amine group forms electrostatic enforced hydrogen bond with E150 and the carbonyl group of T250, the other primary amine group makes hydrogen bonds with the side chain of E150

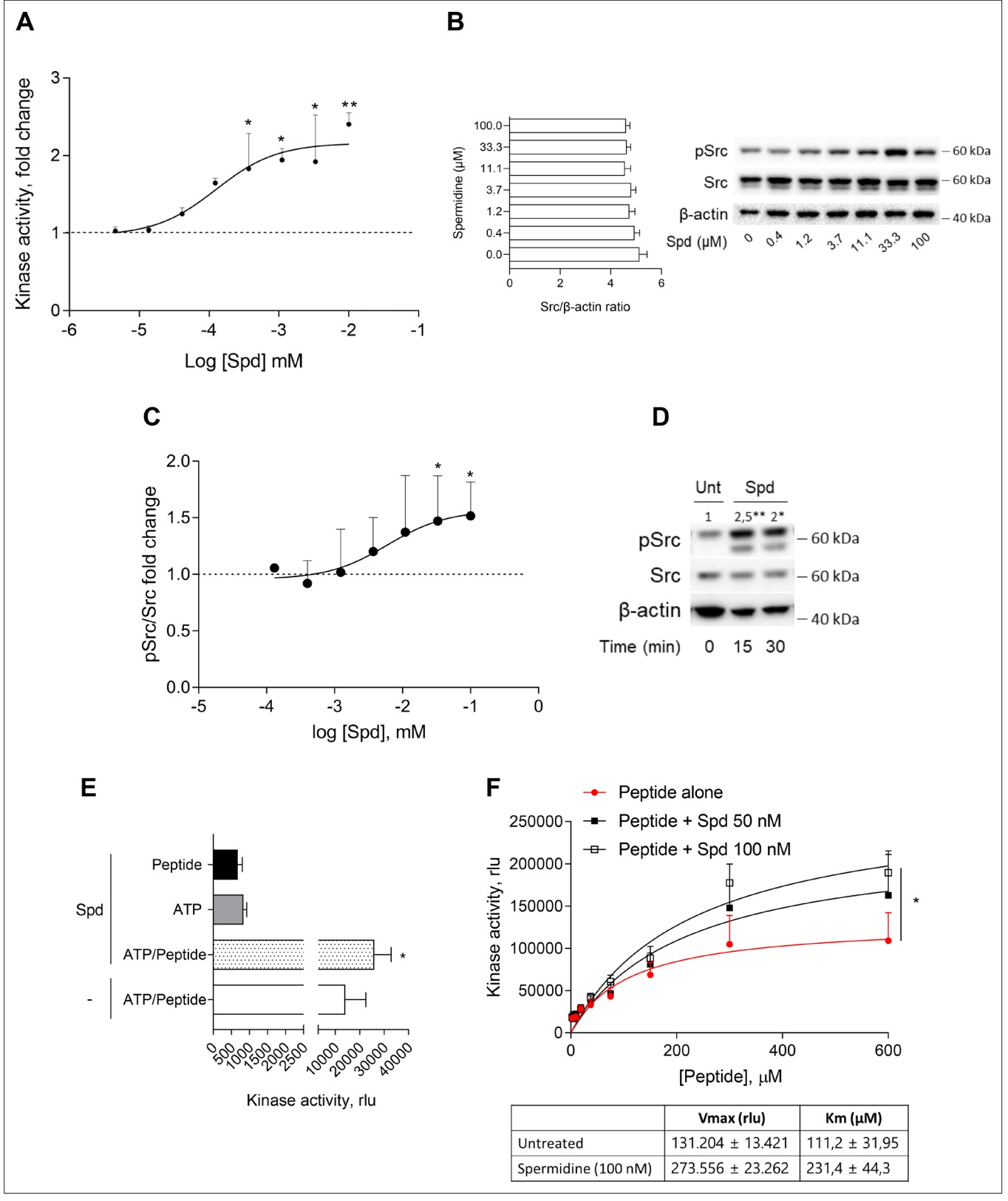

**Figure 1.** Spermidine enhances the activity of Src kinase in ATP-independent manner. (**A**) Enzymatic activity of rhSrc in the presence of ATP (10 μM), synthetic peptide (100 μM), and increasing concentration of spermidine (45 nM to 100 μM). ADP-Glo Kinase Assay (Promega) was used to detect the activity. Results are shown as fold change vs untreated samples (fold change = 1, dotted line). Data are mean ± standard deviation (SD) of three independent experiments, each performed in triplicates. Data were analyzed with one-way analysis of variance (ANOVA) followed by post hoc

*Figure 1 continued on next page*

*Figure 1 continued*

Bonferroni test, by comparing the mean of spermidine-treated samples to untreated counterpart. *p < 0.05, **p < 0.01. Spermidine EC50 = 106.4 ± 13.4 nM. (**B**) Immunoblot analysis of phosphorylated (pSrc) and total Src protein level evaluated in cell lysates from SYF cells reconstituted with vector coding for wild-type Src and then treated with increasing concentration of spermidine (400 nM to100 µM). β-Actin expression was used as normalizer and the Src/β-actin ratio is included as mean ± SD of three independent experiments. One representative immunoblot of three is shown. (**C**) pSrc/Src ratio of scanning densitometry analysis of three independent immunoblots. Data (mean ± SD) are reported as fold change of samples treated with spermidine relative to untreated cells (fold change = 1, dotted line). Data were analyzed with one-way ANOVA followed by post hoc Bonferroni test, by comparing the mean of spermidine-treated samples to the untreated counterpart. *p < 0.05. Spermidine EC50 = 6.4 ± 0.6 µM. (**D**) Immunoblot analysis of phosphorylated (pSrc) and total Src protein level evaluated in cell lysates from MC38 cells treated with spermidine (20 µM) for the indicated time. β-Actin expression was used as normalizer. pSrc/Src ratio is calculated by densitometric quantification of the specific bands and is reported as fold change against untreated cells (fold change = 1). Data were analyzed with one-way ANOVA followed by post hoc Bonferroni test, by comparing the mean of spermidine-treated samples to the untreated counterpart. *p < 0.05, **p < 0.01 (**E**) Enzymatic activity of rhSrc in the presence of spermidine, with or without ATP and peptide substrate. Data are mean ± SD of three independent experiments and were analyzed by Student's *t*-test comparing the Spd/ATP/peptide vs ATP/peptide sample. (**F**) Enzymatic activity of rhSrc in the presence of fixed concentrations of spermidine and increasing concentration of peptide substrate. Data are reported as mean ± SD of three independent experiments, each performed in triplicates. Vmax and Km were calculated after fitting the kinase activity data to the Michaelis–Menten equation. Data were analyzed with one-way ANOVA followed by post hoc Bonferroni test. *p < 0.05.

The online version of this article includes the following source data and figure supplement(s) for figure 1:

**Source data 1.** Original immunoblots of phosphorylated (pSrc), total Src and actin protein levels evaluated in cell lysates from SYF cells reconstituted with vector coding for wild-type Src and then treated with increasing concentration of spermidine.

**Source data 2.** Original immunoblots of phosphorylated (pSrc), total Src and actin protein levels evaluated in cell lysates from MC38 cells either treated with spermidine or left untreated for 15 and 30 min.

**Figure supplement 1.** Spermidine does not compete with ATP and does not potentiate the constitutive active Src.

**Figure supplement 1—source data 1.** Original immunoblots of phosphorylated, total Src and β-tubulin protein levels in lysates from SYF cells either reconstituted with vector coding for wild-type Src or Src mutated at tyrosine 529 with phenylalanine and then exposed to spermidine .

and the carbonyl group of A148 while engaging the aromatic ring of F153 through a specific π-cation interaction (*Macchiarulo et al., 2009*).

To experimentally confirm the proposed spermidine-binding site, we resorted to mutagenesis experiments by substituting the residues E150 or E169, corresponding to E149 and E168 of murine Src sequence into alanine (E149A and E168A). SYF cells were thus stably transfected with vectors coding for the mutated Src (i.e., Src E149A and Src E168A) and wild-type Src (WT) (*Figure 2—figure supplement 2*). To validate the functional equivalence of Src mutants, cells were exposed to lyso-phosphatidic acid (LPA), a stimulus known to activate Src kinase downstream the LPA2 receptor in SYF cells (*Lai et al., 2005*). Results indicated that Src activity is induced by LPA as measured by the phosphorylation of the Y418, independently of the mutation at the putative allosteric site (*Figure 2—figure supplement 2*). On evaluating the activation of Src by spermidine, we found that the mutation of the glutamate residues abrogated the kinase activation (*Figure 2E, F*). The split-luciferase fragment complementation assay confirmed that E149 and E168 are key anchoring points for spermidine binding. Specifically, SYF cells expressing Src WT or mutant were stably transfected with a bioluminescent reporter that contains the SH2 domain and the Src consensus substrate peptide between the amino-(Nluc) and carboxyl-(Cluc) terminal domains of the Firefly luciferase molecule (*Figure 2G*; *Niu and Chen, 2012*). When the endogenous Src is active, the tyrosine residue of the consensus peptide is phosphorylated and interact with the docking pocket of the SH2 domain. This creates a steric hindrance that prevents the reconstitution of a functional luciferase, resulting in a reduction of bioluminescent signal (*Figure 2G*). Cells co-expressing Src and the reporter were thus exposed to spermidine and the luminescent signal was measured. Results demonstrated that the bioluminescence decreased when spermidine is applied only in cells ectopically expressing wild-type Src (*Figure 2H*). Of note, in the absence of spermidine, cells – those reconstituted with the reporter and the different mutants of Src – did not generate statistically different luminescent signal (617.7 ± 121 vs 552.2 ± 68.9 vs 501 ± 110, WT Src vs E149A vs E168A, respectively). This suggests that the basal activity of the kinase is not affected by the specific amino acid substitution.

Overall, these data suggested the presence of a previously unknown allosteric site on the backside of Src SH2 domain as defined by the glutamate residues at positions 150 and 169 of the human sequence. Spermidine, by means of ionic and hydrogen bond interactions between its protonated

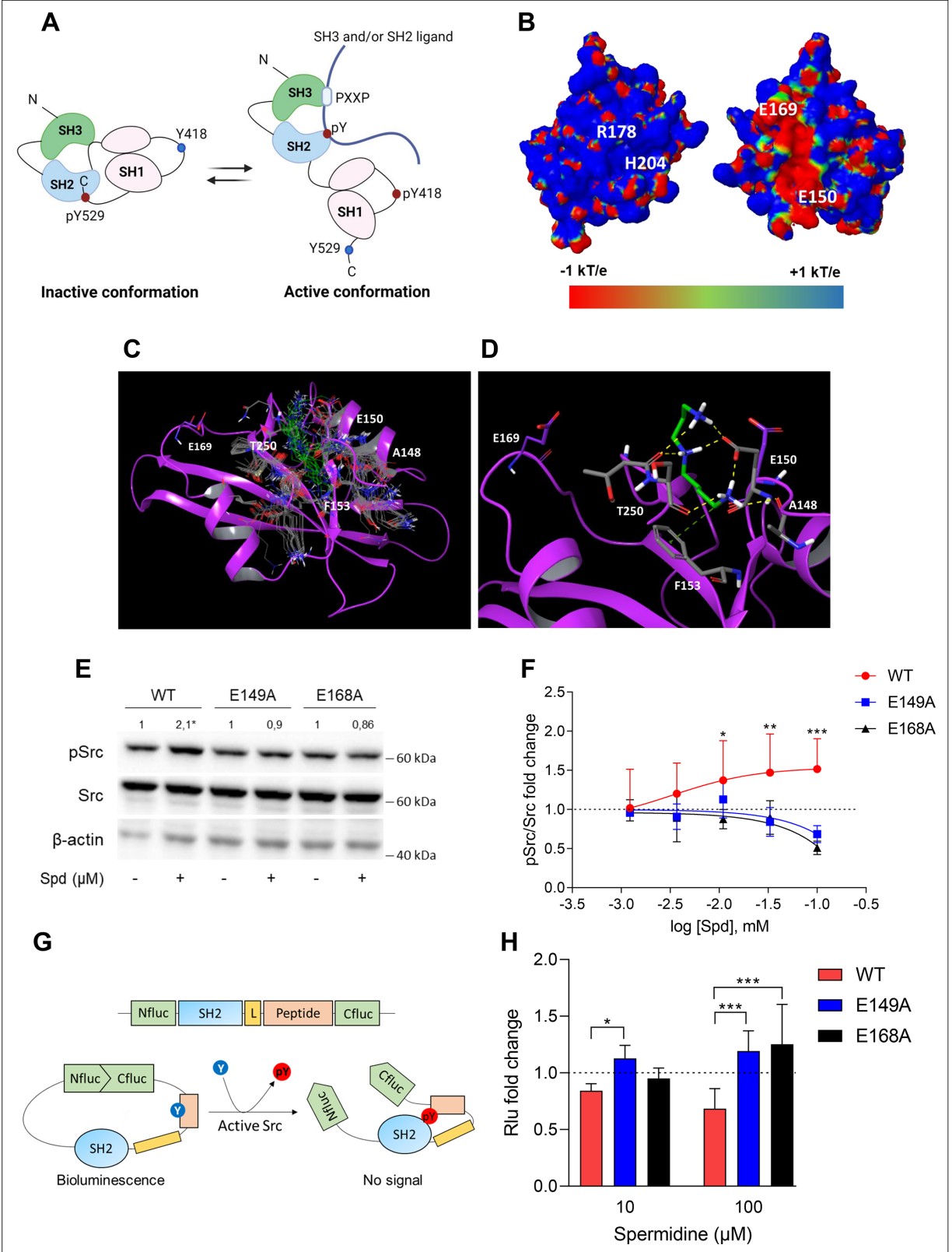

**Figure 2.** Spermidine binds to an allosteric site located in the SH2 domain of Src kinase. (**A**) Schematic representation of the murine Src domains and kinase activation. The catalytic activation of the enzyme is characterized by the phosphorylation of the Y418 (pY418) in the activation loop. Created with BioRender.com. (**B**) Electrostatic potential surface of the Src SH2 domain showing the pY-binding site (R178 and H204) and the putative allosteric site for the endogenous polyamine as delimited by the glutamate residues (E150 and E169). Residues are labeled according to sequence numbering of the

*Figure 2 continued on next page*

*Figure 2 continued*

human isoform: R178, H204, E150, and E169 correspond to R32, H58, E4, and E23 of the NMR structure of the viral isoform, respectively. (**C**) Overlay of docking solutions of spermidine into the shallow cavity of Src kinase (poses #1–18, ***Supplementary file 1***). Induced-fit conformations of side chains of residues shaping the cavity are shown with gray carbon-atoms according to each docking solution. Conformations of spermidine according to each docking solution are shown with green carbon-atoms. The Src SH2 domain is shown with magenta cartoon depicting the secondary structure. Residues are labeled according to sequence numbering of the human isoform. E150 and E169 residues are shown with magenta carbon-atoms. (**D**) Best energy-scored solution of the binding mode of spermidine into the allosteric pocket of Src (pose #1, ***Supplementary file 1***). E149 and E168 are shown with magenta carbon-atoms. Interacting residues and spermidine are shown with gray and green carbon-atoms, respectively. Hydrogen bond interactions are shown with yellow dashed lines, while the π-cation interaction is reported with green dashed line. (**E**) Immunoblot analysis of phosphorylated (pSrc) and total Src protein level in cell lysates from SYF cells either reconstituted with vector coding for wild-type Src (WT) or Src mutated at glutamate 149 or 168 with alanine (E149A and E168A). Cells were then exposed to spermidine (100 µM). β-Actin expression was used as normalizer. pSrc/Src ratio is calculated by densitometric quantification of the specific bands and is reported as fold change against the corresponding untreated cells. Data were analyzed with one-way analysis of variance (ANOVA) followed by post hoc Bonferroni test, by comparing the mean of spermidine-treated samples to the untreated counterpart. *$p < 0.05$. (**F**) Activation of Src kinase in SYF cells treated with increasing concentration of spermidine (400 nM to 100 µM) and measured as pSrc/Src ratio of scanning densitometry analysis of three independent immunoblots. Results (mean ± standard deviation [SD]) are reported as fold change of samples treated with spermidine relative to untreated cells (fold change = 1, dotted line). (**G**) Schematic representation of the reporter functions. In the presence of active Src kinase, the phosphorylation of Src peptide results in its intramolecular interaction with the SH2 domain that prevents the complementation of split-luciferase fragments and generates a reduced bioluminescence activity. In the absence of Src activation, the N- and C-terminal luciferase domains are reconstituted and thus the bioluminescent activity is restored. (**H**) Measurement of luminescent signal in SYF cells co-expressing the reporter and the wild-type Src or its mutants (E149A and E168A), and then exposed to spermidine (10 and 100 µM). Results (mean ± SD of three independent experiments) are reported as fold change of bioluminescent signal in stimulated cells as compared to their respective untreated samples. Data (**F, H**) were analyzed with two-way ANOVA followed by post hoc Bonferroni test. *$p < 0.05$, **$p < 0.01$, ***$p < 0.001$.

The online version of this article includes the following source data and figure supplement(s) for figure 2:

**Source data 1.** Original immunoblots of phosphorylated (pSrc), total Src and actin protein level evaluated in cell lysates from SYF cells either reconstituted with vector coding for wild-type Src (WT) or Src mutated at glutamate 149 or 168 with alanine.

**Figure supplement 1.** The glutamate residues E149 and E168 are conserved across different species.

**Figure supplement 2.** Efficient reconstitution of SYF cells with vectors coding for Src kinase.

**Figure supplement 2—source data 1.** Original immunoblots of phosphorylated (pSrc), total Src and actin protein levels in cell lysates from SYF cells either reconstituted with vector coding for wild-type Src (WT) or Src mutated at glutamate 149 or 168 with alanine.

**Figure supplement 2—source data 2.** Original immunoblots of phosphorylated, total Src and actin protein levels in lysates from SYF cells either reconstituted with vector coding for wild-type Src or Src mutated at glutamate 149 or 168 with alanine.

amino groups and residues of the shallow anionic site on the SH2 domain, directly associates with and activates Src kinase. It is worth noting that no direct interaction was observed between spermidine and E169 in the docking study. This may be ascribed to the limit of the scoring function in identifying a binding mode engaging E169 among resulting solutions, or to an indirect role of such residue in promoting long-range electrostatic interactions to accomplish the molecular recognition of the cognate ligand into the allosteric site. Moreover, the alignment of murine Src, Yes, and Fyn sequences highlighted that only the glutamate residue at position 149 of murine Src is conserved across SH2 domains, while the glutamate residue at position 168 is replaced by the amino acid glycine (data not shown). Thus, although we cannot exclude that spermidine can bind the SH2 domain of related Src kinases, we might speculate that the lack of the negative charge at position 168 reduces the long-range electrostatic interactions that we hypothesized to be responsible for the molecular recognition of the ligand into the allosteric site.

## Spermidine promotes the Src-dependent tyrosine phosphorylation of IDO1 and their interaction

Among the proteins phosphorylated by Src, the immunometabolic enzyme IDO1 is worthy of note (***Mondanelli et al., 2017***; ***Manni et al., 2020***). Indeed, aside metabolizing the amino acid tryptophan, IDO1 is endowed with non-enzymatic properties (***Mondanelli et al., 2020a***; ***Albini et al., 2017***; ***Pallotta et al., 2011***; ***Orabona et al., 2012***; ***Orecchini et al., 2023***; ***Albini et al., 2018***). The latter relies on the presence of two ITIMs (immunoreceptor-tyrosine-based inhibitory motif) that can be phosphorylated in response to immunomodulatory stimuli, such as TGF(transforming growth factor)-β, L-kynurenine, and spermidine (***Mondanelli et al., 2017***; ***Manni et al., 2020***; ***Pallotta et al., 2011***). However, the exact molecular mechanism and the role of spermidine have never been explored. To

confirm that Src can phosphorylate IDO1, SYF cells were reconstituted with vectors coding for murine wild-type Src and IDO1, either alone or in combination, and then were exposed to spermidine. Results from immunoblot demonstrated that IDO1 phosphorylation increased when Src is co-expressed and activated by spermidine (*Figure 3A*). In addition, we found that in the presence of a constitutive active Src (i.e., Src Y529F) IDO1 phosphorylation increases regardless of the stimulation with spermidine (*Figure 3—figure supplement 1*). This suggests that spermidine can act as an on/off switcher of the kinase, without potentiating the constitutively active protein.

To further confirm that spermidine could promote the IDO1 phosphorylation by accelerating the reaction velocity, an in vitro kinase assay was performed using purified human Src and IDO1 proteins. By detecting phosphotyrosine residues with a specific antibody, we found that IDO1 was phosphorylated in a time-dependent manner (*Figure 3B, C*). Moreover, in the presence of spermidine, Src quicker phosphorylated IDO1, as demonstrated by the twofold increase of the relative velocity (*Figure 3D*). To figure out whether the IDO1 phosphorylation was a direct effect through physical interaction with Src, SYF cells reconstituted with murine wild-type Src and IDO1 were exposed to spermidine for different length of time. Co-immunoprecipitation followed by immunoblot studies demonstrated that when cells were treated with spermidine for 60 min, IDO1 was found in a complex with Src (*Figure 3E*). Of note, among the predicted functional partners of murine Src, we verified additional substrates (i.e., Stat3) and we found that the spermidine effect is specific to the IDO1–Src complex (data not shown). The specific IDO1–Src interaction was confirmed in situ by the proximity ligation assay (*Figure 3F, G*). Accordingly, spermidine treatment induced the Src-IDO1 interaction in SYF cells reconstituted with wild-type Src, but not with the E149A or E168A mutant form of the kinase (*Figure 3F, G*). To prove a physiological relevance of this interaction, the spermidine effect was also evaluated in MC38 cells endogenously expressing the proteins IDO1 and Src. Results demonstrated that when MC38 cells are exposed to spermidine, the phosphorylation of IDO1 increased (*Figure 3H*) as well as the IDO1/Src interaction (*Figure 3I*).

As a whole, these results suggested that spermidine not only accelerates the Src-mediated phosphorylation of IDO1, but also promotes the formation of Src–IDO1 complex.

## Discussion

The non-receptor tyrosine kinase Src is the representative of a family of structure-related enzymes involved in several signaling pathways regulating key cellular processes as well as immune responses (*Caner et al., 2021*; *Liu et al., 2013*). Much relevant literature correlates dysregulated Src kinase activity with cancer and thus extensive efforts have been made to develop small molecules kinase inhibitors. Currently, approved kinase inhibitors are compounds that reversibly bind the catalytic site and thus compete with the ligand (i.e., ATP) (*Roskoski, 2015*). As the ATP-binding cleft is structurally well conserved among kinases, these inhibitors are poorly selective. Moreover, their chronic usage is frequently associated with acquired drug resistance that ultimately limits patients' compliance and the therapeutic success. For instance, the FDA-approved Dasatinib and Bosutinib inhibit more than 30 kinases, and thus are not suitable for probing Src-dependent pharmacology (*Ozanne et al., 2015*). Saracatinib is another example of small molecule that interacts with the ATP-binding pocket. Although more selective than Dasatinib, it potently inhibits EGF-R as well (*Formisano et al., 2015*). In addition to competitive Src inhibitors, an emerging pharmacological modality – known as targeted covalent inhibitors – has been pursued at the preclinical level for blocking Src kinase activity (*Gurbani et al., 2020*). However, the promiscuity of molecules interacting with the ATP pocket has moved the interest toward the development of alternative strategies for more effective and less-toxic inhibitors.

The peculiarity of the Src protein, as well as of other tyrosine kinases, is its structural plasticity, that is, the capability to adopt distinct conformations due to intrinsic dynamic properties (*Engen et al., 2008*). The activation state of this protein kinase is indeed dictated by dynamic intramolecular interactions between the SH3, SH2, and kinase domains. The SH2 domain plays a key role in both autoregulating Src kinase activity and in recruiting the protein ligand. Specifically, the tyrosine residue at position +530, when phosphorylated, interacts with the SH2 domain and stabilizes a restrained catalytically inactive conformation of Src (*Shah et al., 2018*). Accordingly, binding of ligand proteins to the SH2 domain displaces intramolecular contacts and promotes the catalytic activation of the Src kinase. This activating event is mostly driven by a dynamic breakage and formation of electrostatic interactions that involve salt bridges and hydrogen bonds. Guided by the dynamic nature of the

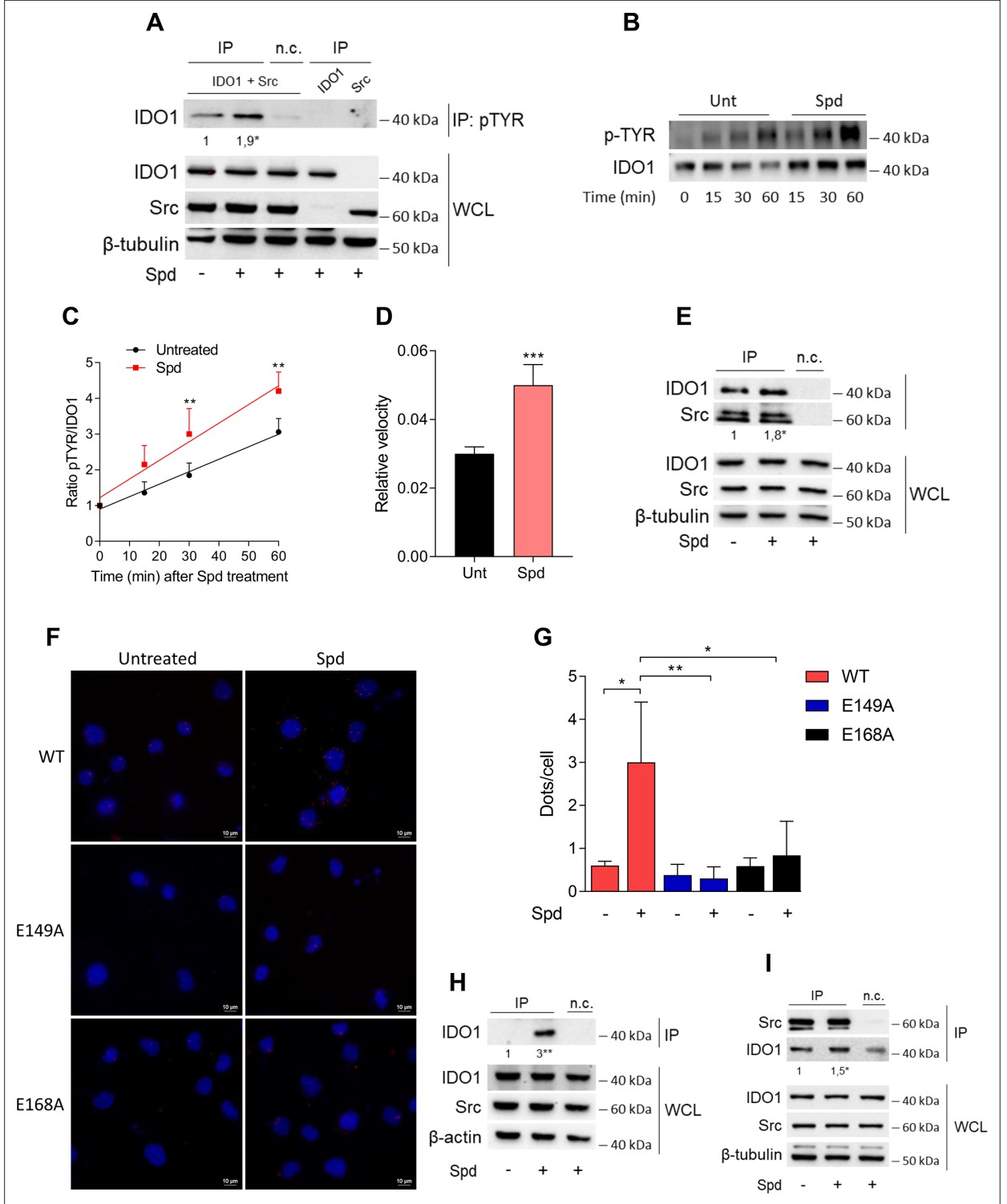

**Figure 3.** Spermidine triggers the phosphorylation of indoleamine 2,3-dioxygenase 1 (IDO1) by Src kinase and the complex formation. (**A**) Immunoprecipitation with anti-phosphotyrosine antibody from SYF cells reconstituted with vectors coding for Src and IDO1 and then treated with spermidine (100 µM) for 60 min. Cells transfected with vectors coding for either Src or IDO1 were used as control, while immunoprecipitation without antibody was included as negative control (n.c.). The detection of IDO1, Src, and β-tubulin was performed by sequential immunoblotting with specific antibodies. Whole-cell lysates (WCL) was used as control of protein expression. One representative immunoblot of three is shown. The amount of IDO1 precipitated is measured by densitometric quantification of the specific bands in treated sample co-expressing IDO1 and Src and is reported relative to untreated cells (fold change = 1). Data (mean of three experiments) were analyzed with unpaired Student's *t*-test. *p < 0.05. (**B**) Continuous in vitro

*Figure 3 continued on next page*

*Figure 3 continued*

kinase assay with rhIDO1 (300 ng) and rhSrc (50 ng) followed by immunoblot analysis with anti-phosphotyrosine and anti-IDO1 specific antibodies. The reaction was carried out for the indicated time, in either the presence or absence of spermidine. One representative immunoblot of three is shown. (**C**) pTYR/IDO1 signals were calculated by densitometric quantification of the specific bands. Data were plotted over incubation time of the kinase reaction and the slopes (relative velocity) of linear fits were calculated. Results (mean ± standard deviation [SD]) were analyzed with two-way analysis of variance (ANOVA) followed by post hoc Bonferroni test and by comparing, for each time point, the pTYR/IDO1 ratio of spermidine-treated sample to the untreated counterpart. (**D**) The relative velocity of the kinase reaction in either the presence or absence of spermidine from three independent experiments is shown. Data (mean ± SD) were analyzed with unpaired Student's *t*-test. ***p < 0.001. (**E**) Immunoprecipitation of Src from SYF cells reconstituted with Src and IDO1, and then treated as in (**A**). The detection of IDO1, Src, and β-tubulin was performed by sequential immunoblotting with specific antibodies. Immunoprecipitation without antibody was included as negative control (n.c.). Whole-cell lysates (WCL) was used as control of protein expression. One representative immunoblot of three is shown. IDO1/Src ratio is calculated by densitometric quantification of the specific bands and is reported as fold change against untreated cells. Data (mean of three independent experiments) were analyzed with unpaired Student's *t*-test. *p < 0.05. (**F**) The in situ proximity ligation assay between IDO1 and Src in SYF cells reconstituted with wild-type Src or the mutant forms and treated as in (**A**). Red spots indicate a single IDO1/Src interaction; scale bars, 10 µm. One representative experiment of three is shown. (**G**) Quantification of the interactions detected by proximity ligation assay using ImageJ. Results are reported as function of the number of cells. Data (mean ± SD) were analyzed with one-way ANOVA followed by post hoc Bonferroni test. *p < 0.05, **p < 0.01. (**H**) Immunoprecipitation with anti-phosphotyrosine antibody from MC38 cells either treated with spermidine (20 µM) for 60 min or left untreated. Immunoprecipitation without antibody was included as negative control (n.c.). The detection of IDO1, Src, and β-actin was performed by sequential immunoblotting with specific antibodies. Whole-cell lysates (WCL) was used as control of protein expression. One representative immunoblot of three is shown. The amount of IDO1 precipitated is measured by densitometric quantification of the specific band and is expressed relative to untreated cells (fold change = 1). Data (mean of three independent experiments) were analyzed with unpaired Student's *t*-test. **p < 0.01. (**I**) Immunoprecipitation of Src from MC38 cells treated as in (**H**). The detection of IDO1, Src, and β-tubulin was performed by sequential immunoblotting with specific antibodies. Whole-cell lysates (WCL) was used as control of protein expression. One representative immunoblot of three is shown. IDO1/Src ratio is calculated by densitometric quantification of the specific bands and is reported as fold change against untreated cells (fold change = 1). Data (mean of three independent experiments) were analyzed with unpaired Student's *t*-test. *p < 0.05.

The online version of this article includes the following source data and figure supplement(s) for figure 3:

**Source data 1.** Original immunoblots of immunoprecipitation with anti-phosphotyrosine antibody (IP) followed by the detection of indoleamine 2,3-dioxygenase 1 (IDO1) with specific antibodies.

**Source data 2.** Original immunoblots of in vitro kinase assay with rhIDO1 (300 ng) and rhSrc (50 ng) followed by immunoblot analysis with anti-phosphotyrosine and anti-IDO1 specific antibodies.

**Source data 3.** Original immunoblots of immunoprecipitation with anti-phosphotyrosine antibody (IP) followed by the detection of indoleamine 2,3-dioxygenase 1 (IDO1) with specific antibodies.

**Source data 4.** Original immunoblots of immunoprecipitation of Src from MC38 cells treated with spermidine or left untreated.

**Source data 5.** Original immunoblots of immunoprecipitation of Src from MC38 cells treated with spermidine or left untreated.

**Figure supplement 1.** Spermidine does not promote the phosphorylation of indoleamine 2,3-dioxygenase 1 (IDO1) via constitutive active Src.

**Figure supplement 1—source data 1.** Original immunoblots of immunoprecipitation with anti-phosphotyrosine antibody (IP) followed by the detection of indoleamine 2,3-dioxygenase 1 (IDO1) with specific antibodies.

kinase, allosteric modulation has been proposed as pharmacological approach to target the activity of Src kinase. Allosteric molecules do not possess intrinsic efficacy, but instead modulate – either positively or negatively – the activity of orthosteric agonists. Moreover, being less conserved among kinases, the allosteric hotspots ensure greater drug selectivity. Targetable allosteric pockets have been identified for few kinases as reported for Hck, Lyn, Aurora A kinase, and Bcr-Abl (*Dorman et al., 2019*; *Zhang et al., 2010*; *Saporito et al., 2012*; *Panicker et al., 2019*). In addition, modulators of the SH2 and SH3 domains – either peptidomimetics or small molecules – have been developed as chemical tools modifying the conformation and thus both the enzymatic and non-enzymatic functions of Src, the latter including protein–protein interactions and intracellular localization (*Moroco et al., 2015*; *Moroco et al., 2014*; *Saporito et al., 2012*; *Leonard et al., 2014*; *Fischer et al., 2015*).

Metabolites are chemicals that do not merely take place in the metabolic reactions, but are also involved in inter- and intracellular communications, energy production, macromolecule synthesis, post-translational modifications, and cell survival (*Gargaro et al., 2022*; *Makowski et al., 2020*; *Icard et al., 2019*; *Diskin et al., 2021*; *Mondanelli and Volpi, 2021*; *Mondanelli and Volpi, 2020*). In accordance, the enzymes responsible for their production are considered central regulators of the function of cells, including immune cells (*Giovanelli et al., 2019*; *Grohmann et al., 2017*; *Sun et al., 2022*; *Mandarano et al., 2020*; *Bonometti et al., 2023*). IDO1 is the prototype of such metabolic enzymes acting at the forefront of immune responses. Thanks to its catalytic activity as well as non-enzymatic

function (relying on the phosphorylation of its ITIMs), IDO1 is a tiebreaker of tolerance and immunity (*Grohmann et al., 2017*; *Mondanelli et al., 2019*; *Iacono et al., 2020*). Prompted by the finding that spermidine (i.e., a natural occurring polyamine) can reprogram murine cDCs toward an immunoregulatory phenotype *via* the Src kinase-dependent induction of the IDO1 signaling (*Mondanelli et al., 2017*), we here demonstrated that the polyamine behaves as a positive allosteric modulator of Src by increasing the maximum rate of enzyme activity. Indeed, electrostatic potential calculation studies on the SH2 domain identified a surface endowed with a negative electrostatic potential on the back side of the pY-binding site, as delimited by glutamate residues E149 and E168. By its protonated amino groups, spermidine interacts with the anionic head of E149 on the SH2 domain, directly associates with and activates Src kinase. As a matter of fact, the site-directed mutagenesis of the glutamate residues with uncharged amino acids abrogates the spermidine-mediated activation of Src kinase. On note, by aligning murine Src with several SH2 domain proteins (*Liu et al., 2011*) such as enzymes (Abl1, SHIP1), docking (Shc1), and adaptor (Grb2, SOCS1) proteins, we found that neither E149 nor E168 are conserved, in their place there are non-polar hydrophobic amino acids (e.g., alanine, glycine, and proline) or those with polar uncharged side chains (e.g., glutamine) (data not shown). These would suggest that spermidine cannot broadly bind SH2-domain containing proteins and that the polyamine selectively affects the Src kinase activity. It is noteworthy of mention that polyamines are not new in the field of allosteric modulation, as they modify the activity of ionotropic *N*-methyl-D-aspartate receptor (NMDAR, a receptor for glutamate) by both increasing the affinity of NMDAR for the co-agonist glycine and relieving the tonic proton inhibition of the receptor (*Hirose et al., 2015*), further supporting the spermidine mode of action. Polyamines are usually considered as a family of molecules with similar functions; however, different polyamines may have distinct, sometimes opposite, roles as in inflammatory- and age-related pathologies, (*Minois et al., 2011*). Concordantly, they have a different regulatory capacity on Src. For instance, putrescine and spermidine, but not spermine, promote the phosphorylation of Src in tumor cells (*Hölttä et al., 1993*), while spermine negatively modulates the activity of the kinase in intestinal epithelial cells (*Ray et al., 2012*). In cDCs, putrescine and spermidine, but not spermine, increased Src phosphorylation and when compared the IDO1-inducing ability

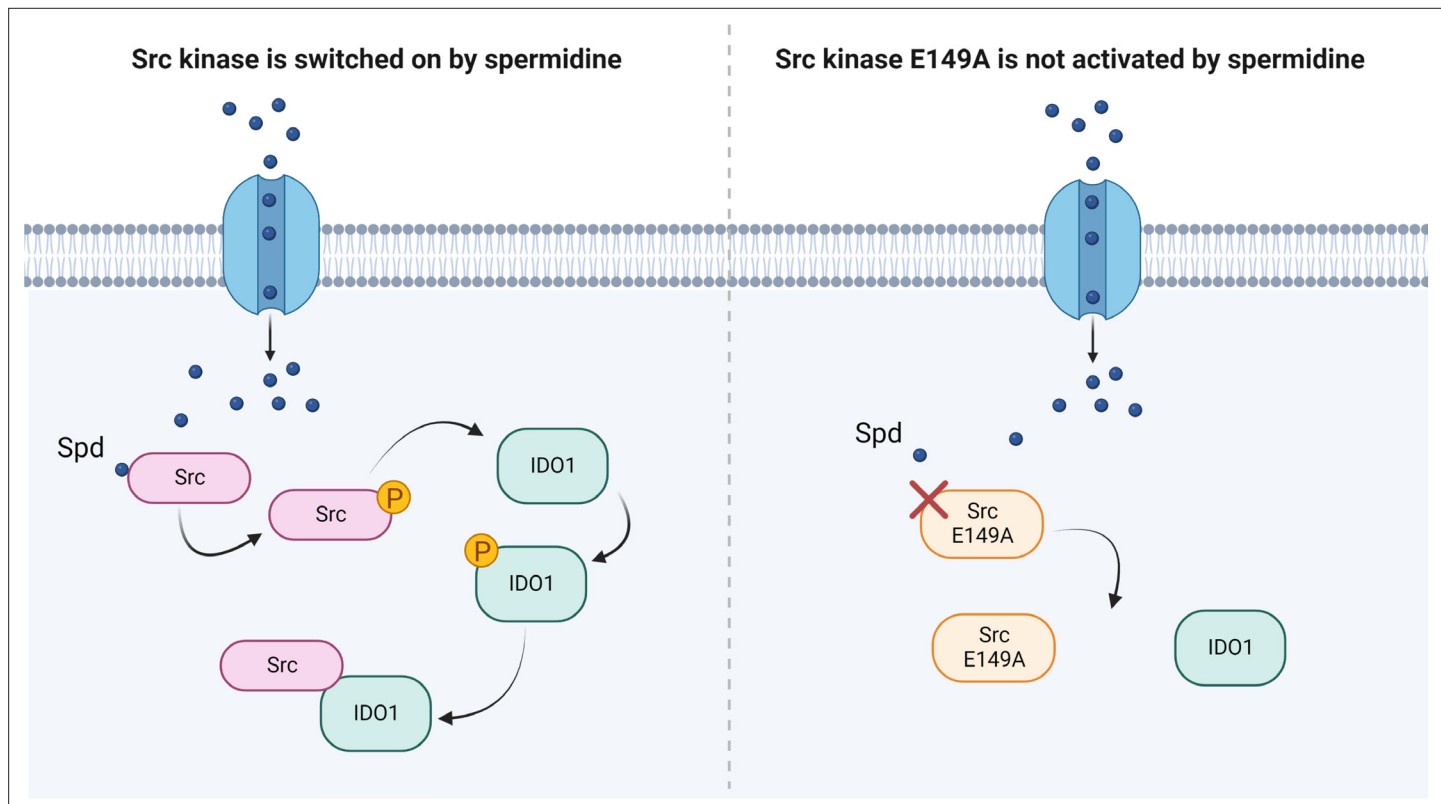

**Figure 4.** Scheme of the Src kinase modulation by the polyamine spermidine. Created with BioRender.com.

**Table 1.** Primers for site-directed mutagenesis of Src.

| Primer | Sequence |
| --- | --- |
| Src E149A, *Forward* | atccaggctgaggcgtggtacttt |
| Src E149A, *Reverse* | aaagtaccacgcctcagcctggat |
| Src E168A, *Forward* | ctcaacgccgcgaacccgaga |
| Src E168A, *Reverse* | tctcgggttcgcggcgttgag |
| Src Y529F, *Forward* | gagccacagttccagcccgg |
| Src Y529F, *Reverse* | ccgggctggaactgtggctc |

of polyamines, the induction of the IDO1 protein could be observed only for spermidine-treated cells (*Mondanelli et al., 2017*).

Besides confirming that Src phosphorylates IDO1, and that the polyamine accelerates the enzyme kinetic, here we showed that spermidine promotes the interaction of Src with IDO1 protein (*Figure 4*). Our data provided evidence that an endogenous metabolite, when present at specific concentrations, can directly activate Src kinase without requiring a membrane receptor. By acting on the backside of the SH2 domain, that is the domain responsible for the substrate binding, spermidine not only modulates the catalytic activity, but also affects the scaffold function of Src in organizing transducing signaling complexes – as those with IDO1 – which could be relevant in many diseases. Thus, from a therapeutic perspective, our results provide the proof of principle for the development of molecules that can modulate the kinase activity and the non-enzymatic functions of Src and IDO1 at once.

## Materials and methods
### Cell lines and reagents
SYF cells that is fibroblast null for Src, Fyn, and Yes kinases (*Klinghoffer et al., 1999*); RRID:CVCL_6461 were grown in DMEM(Dulbecco's Modified Eagle Medium) supplemented with 10% FCS(fetal calf serum), at 37°C, in a humidified 5% $CO_2$ incubator. SYF cells were purchased by ATCC (# CRL-2459). MC38 cells were kindly provided by Stefano Ugel (University of Verona, Italy) and were cultured as indicated for SYF cells. Both cell lines were tested to confirm the absence of mycoplasma contamination by N-GRADE Mycolpasma PCR Reagent set (#EMK090020). Spermidine, LPA, and recombinant Src protein were purchased from Sigma-Aldrich, while recombinant human IDO1 protein was obtained by Giotto Biotech. Construct (pCMV6-Entry) expressing murine Src was obtained from Origene (MR227248). Src was then subcloned into pEF.bos plasmid by using degenerate primers bearing the unique restriction site (namely Kpn I and Not I). Src mutants were generated by PCR(Polymerase chain reaction)-based site-directed mutagenesis performed with overlap extension that involved mutagenic primers (*Table 1*) in two independent PCRs before combining them in the final PCR (*Reikofski and Tao, 1992*; *Ho et al., 1989*). The resulting PCR products were digested with appropriate restriction enzymes and cloned into pEF-BOS plasmid.

### Cell transfection and treatment
SYF cells (0.3 millions/ml in 6-well plate) were plated the day before and were transfected with 2 µg of the vectors expressing either wild-type Src or Src mutants according to the Lipofectamine 3000 protocol (Thermo Fisher Scientific). Stable transfectants were obtained by antibiotic selection (2.5 µg/ml) of SYF cells transfected with pEF-BOS-based vectors carrying the puromycin resistant genes. SYF and MC38 cells were serum starved overnight before spermidine treatment. Cells were incubated with the polyamine or LPA for 60 min for immunoblot analysis, as otherwise indicated. These conditions were selected based upon optimization experiments.

### Split-luciferase fragment complementation assay
The N and C fragments of luciferase were amplified by PCR from pGL3-Basic. The fragment including nucleotide sequences of SH2 domain of murine Src (aa 374–465), linker (SRGGSTSGSGKPGSGEGSG), and Src consensus substrate peptide (WMEDYDYVHLQG), was synthesized by sequential reactions of PCR amplification. This cassette and the luciferase fragments were cloned into pCDNA 3.1 vector.

SYF cells stably expressing the reporter were transfected with wild-type Src and Src E149A or Src E168A and then cultured into 96-well plate, in serum-free medium. After treatment with spermidine

for 2 hr, cells were washed with PBS1X and lysed with PLB-lysis buffer. Luciferase activity was measured with the Luciferase Reporter Assay Kit (Promega).

## Immunoblot and co-immunoprecipitation studies

For immunoblotting, proteins were extracted in M-PER buffer (Thermo Fisher Scientific) supplemented with phosphatases and proteases inhibitors cocktails (Thermo Fisher Scientific) and run on sodium dodecyl sulfate–polyacrylamide gel electrophoresis (SDS/PAGE). The pSrc/Src ratio was assessed with a rabbit Phospho-Src Family (Tyr416) Antibody (#2101, Cell Signaling Technology, Danvers, MA, USA; RRID:AB_331697), recognizing the phosphorylation at tyrosine 418 in murine Src, followed by the detection of total Src by rabbit monoclonal antibody (36D10, Cell Signaling Technology, Danvers, MA, USA; RRID:AB_2106059), as previously shown (*Manni et al., 2020*). Mouse monoclonal antibody against β-actin (Sigma-Aldrich; RRID:AB_262137) was used as normalizer.

Co-immunoprecipitation appraises were performed following the manufacturer's protocol (Thermo Fisher) and as previously shown (*Gargaro et al., 2022*). Briefly, lysates were incubated overnight at 4°C with Dynabeads Protein G, prepared by blocking 12.5 µl of magnetic beads with PBS1X containing 0.5% bovine serum albumin (wt/vol) and bound to 2.5 µg of rabbit anti-Src (36D10) or MultiMab Rabbit Phospho-Tyrosine (P-Tyr-1000; RRID:AB_2687925) antibody. After washing with buffer (25 mM citric acid, 50 mM Dibasic Sodium Phosphate dodecahydrate pH 5), the immuno-complex was eluted with Elution buffer (0.1 M Sodium Citrate dihydate pH 2–3) and Laemmli buffer. Proteins were run on SDS–PAGE and the expression of IDO1 and Src were analyzed with a mouse anti-IDO1 antibody (clone 8G-11, Merck) and a rabbit anti-Src monoclonal antibody (36D10). Mouse monoclonal Ab against β-tubulin (Sigma-Aldrich; RRID:AB_2827403) was used as normalizer. Protein expression was measured by using Image Lab software (Bio-Rad) and the densitometric analysis of the specific signals was performed as previously described (*Mondanelli et al., 2020a*).

## Biochemical assay

For the in vitro cell-free assay, 5 ng of recombinant hSrc were combined with 10 µM of ATP and 100 µM of synthetic peptide (KVEKIGEGTYGVVYK) corresponding to amino acids 6–20 of p34cdc2. The reaction was carried out in a buffer containing 100 mM of Tris–HCl (pH 7.2), 125 mM of $MgCl_2$, 25 mM of $MnCl_2$, 250 µM of $Na_3VO_4$, and 2 mM of DTT(dithiothreitol). The mixture was incubated at 25°C for 30 min and the production of ADP was measured using the ADP-Glo kinase assay kit (Promega). Vmax was calculated after fitting the kinase activity data to the Michaelis–Menten equation. For the continuous in vitro assay, 50 ng of recombinant hSrc were incubated in the assay buffer with 300 ng of recombinant hIDO1, 100 µM of ATP, with or without spermidine (50 nM). The reaction was carried out at 25°C for the indicated time and then stopped by the addition of Laemmli buffer. Samples were run on SDS/PAGE and analyzed for the expression of Phospho-Tyrosine and IDO1 using an anti-pTyr-1000 and anti-IDO1 (clone 10.1, Merck) antibodies, respectively.

## Proximity ligation assay

SYF cells expressing WT Src or Src E149A or Src E168A were serum starved, stimulated with spermidine, fixed for 20 min with 4% PFA(Paraformaldehyde), permeabilized for 10 min with Triton-X 0.1% in PBS1X and then blocked. Duolink Proximity Ligation Assay (#DUO92008, Sigma-Aldrich) was performed according to the manufacturer's protocol. Briefly, primary antibodies rabbit a-mouse Src (Thermo Fisher, 7G6M9) and mouse a-mouse IDO1 (clone 8G11, Merk) were conjugated with either PLUS (#DUO92009, Sigma-Aldrich) or MINUS (#DUO92010, Sigma-Aldrich) oligonucleotides to create proximity ligation assay probes. Samples were incubated overnight at 4°C and, subsequently, ligase solution was added for 30 min. The signal was amplified with amplification polymerase solution at 37°C for 100 min. Nuclei were counterstained with 4′,6′-diamidino-2-phenylindole (#DUO82040, Sigma-Aldrich). A total of seven images (on average of 60 cells) per samples were taken with a Nikon inverted microscope (×60 magnification) and analyzed with the software ImageJ.

## Electrostatic potential calculation study and docking study

The NMR structure of the SH2 domain of Src kinase of Rous sarcoma virus (PDB ID: 2JYQ) (*Taylor et al., 2008*) was taken from the protein data bank (https://www.rcsb.org) (*Berman et al., 2000*). It should be mentioned that the SH2 domain of the viral Src kinase shares a high sequence identity (97%)

with the relative SH2 domains of human and murine Src tyrosine kinase (*Figure 2—figure supplement 1*). Hence, it is possible to use the NMR structure of the viral isoform with very good approximation to infer about the electrostatic potential map of human SH2 domain and to investigate the putative binding site of spermidine. The use of the NMR structure of the single viral SH2 domain avoids the bias that a full-length structure of human Src kinase might bring into more time-consuming calculations due to considering multiple conformations between the regulatory and kinase domains of the enzyme (*Figure 2A*). Indeed, diverse conformations of Src may differently affect the electrostatic potential of the SH2 domain as well as potentially mask a putative binding site for spermidine. Atomic coordinates of the viral SH2 domain (PDB ID: 2JYQ) were processed using the program PDB2PQR (*Dolinsky et al., 2007*; *Dolinsky et al., 2004*). The Adaptive Poisson-Boltzmann Solver (APBS) was applied to calculate the electrostatic potential of SH2 domain and map it on the excluded solvent surface (*Baker et al., 2001*). Specifically, the PARSE force field was employed with default parameters including a solute dielectric value = 2, solvent dielectric value = 78.54, solvent probe radius = 1.4 Å, and temperature = 298.150°K. Two calculations were performed using cubic spline charge discretization and a grid dimension of 129 × 129 × 129 Å. The first run adopted a grid spacing of 0.574 × 0.557 × 0.509 Å, for a grid length of 73.433 × 71.279 × 65.163 Å centered at points 2.638 (*x*), 0.458 (*y*), and 0.467 (*z*). The second run used a grid spacing of 0.494 × 0.484 × 0.456 Å for a grid length of 63.196 × 61.929 × 58.331 Å centered at the same point of the first run.

The chemical structure of spermidine was taken from PubChem compound (*Kim et al., 2016*). The structure was processed using LigPrep (Schrödinger Release 2021-3: LigPrep, LLC, New York, NY, 2021) and applying the default settings.

The structure of the SH2 domain of Src kinase (PDB ID: 2JYQ) was also processed for docking calculations employing the Protein Preparation Wizard (PPW) tool, as implemented in Maestro (Schrödinger Release 2021-3: Maestro, Schrödinger, LLC, New York, NY). In particular, hydrogen atoms were added and the internal geometries of the protein were optimized with a coordinate displacement restrain on heavy atoms set to 0.3 Å. The docking study was carried out defining a grid box for calculations centered on the center of mass of residues E4 and E23 (E150 and E169 according to sequence numbering of the human isoform; E149 and E168 according to sequence numbering of the murine isoform). The inner box was sized 10 × 10 × 10 Å. Since the allosteric site features a shallow surface, a ligand induced-fit approach was used to investigate the binding mode of spermidine. Accordingly, docking solutions were obtained using the induced-fit docking algorithm (Schrödinger Release 2021-3: Induced Fit Docking, Schrödinger, LLC, New York, NY, 2021) and the standard protocol to generate up to n.20 binding poses of spermidine into the allosteric site. During calculations, ligand and receptor van der Waals scaling factors were set to 0.5 kcal/mol, respectively. The side chain conformations of residues within 5 Å of the ligand-binding pose were sampled and refined using the OPLS 2005 force field. The structure of spermidine was then redocked with glide and standard precision scoring function into different obtained conformations of the allosteric site, using up to n.20 top energy conformations of the binding site within 30 kcal/mol of the minimum energy conformation.

## Acknowledgements

This research was funded by University of Perugia, Ricerca di base 2019 (RBGMON19; to GM), Associazione Italiana per la Ricerca sul Cancro (AIRC 2019-23084; to CV), Italian Ministry of Education, University, and Research (PRIN 2020L45ZW; to CO), and University of Perugia, Ricerca di base 2020 (INTEGRATE; to AM).

## Additional information

### Funding

| Funder | Grant reference number | Author |
|---|---|---|
| Università degli Studi di Perugia | Ricerca di base 2019 | Giada Mondanelli |

| Funder | Grant reference number | Author |
|---|---|---|
| Associazione Italiana per la Ricerca sul Cancro | AIRC 2019-23084 | Claudia Volpi |
| Italian Ministry of Education, University, and Research | PRIN 2020L45ZW | Ciriana Orabona |
| Università degli Studi di Perugia | Ricerca di base 2020 | Antonio Macchiarulo |

The funders had no role in study design, data collection, and interpretation, or the decision to submit the work for publication.

## Author contributions

Sofia Rossini, Conceptualization, Methodology, Data curation; Marco Gargaro, Conceptualization, Methodology; Giulia Scalisi, Elisa Bianconi, Sara Ambrosino, Eleonora Panfili, Claudia Volpi, Methodology; Ciriana Orabona, Francesca Fallarino, Supervision; Antonio Macchiarulo, Supervision, Methodology; Giada Mondanelli, Conceptualization, Methodology, Writing - original draft, Data curation

## Author ORCIDs

Elisa Bianconi ⬤ http://orcid.org/0000-0003-4597-8056
Giada Mondanelli ⬤ http://orcid.org/0000-0002-0798-0465

## Decision letter and Author response

Decision letter https://doi.org/10.7554/eLife.85872.sa1
Author response https://doi.org/10.7554/eLife.85872.sa2

---

# Additional files

## Supplementary files

• Supplementary file 1. Solutions of the docking study of spermidine into the allosteric site of Src SH2 domain using the structure of the viral isoform (PDB ID: 2JYQ).

• MDAR checklist

## Data availability

All data generated or analyzed during this study are included in the manuscript and supporting file. Figure 1 - Source Data 1; Figure 1 - Source Data 2; Figure 1 - Figure supplement 1 - Source Data 1; Figure 2 - Source Data 1; Figure 2 - Figure supplement 2 - Source Data 1; Figure 2 - Figure supplement 2 - Source Data 2; Figure 3 - Source Data 1; Figure 3 - Source Data 2; Figure 3 - Source Data 3; Figure 3 - Source Data 14; Figure 3 - Source Data 5; Figure 3 - Figure supplement 1 - Source Data 1: contain the original blots used to generate the figures.

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
