## [Editor Report]

This is an important study describing the mechanism of Spermidine modulation of Src kinase, identifying the interacting amino acids and the effect on indoleamine 2,3-dioxygenase 1 (IDO1) activation based on solid evidence. Considering the important role of IDO1 in the immune response this study could provide important information for the design of allosteric modulators capable of turning SRC on/off.

---

## [Decision Letter]

**Decision letter after peer review:**

Thank you for submitting your article "A Back-Door Insights into the modulation of Src kinase activity by the polyamine spermidine" for consideration by *eLife*. Your article has been reviewed by 3 peer reviewers, and the evaluation has been overseen by a Reviewing Editor and Volker Dötsch as the Senior Editor. The reviewers have opted to remain anonymous.

Essential revisions:

1. Please be more clear in the numbering of amino acids and specific species used:

– Line 65. In the introduction, the authors stated "The autoregulatory function of the kinase occurs through intramolecular interactions that stabilize the catalytically inactive conformation of Src, in which the SH2 domain binds to a pY located at position +535." What Src isoform and species are the authors referring to? cSrc is regulated by phosphorylation of Tyr530 in human, Tyr529 in mouse, Tyr527 in avian protein. Providing a reference would be helpful.

– Line 88. The authors stated that they check the phosphorylation of human Src on Tyr424 as an indicator of Src activity. Human Src is phosphorylated on Tyr419, the mouse is on 418, and avian is on 416.

– in vitro experiments were done with human Src but there is no clear statement of what Src construct was used in SYF cells.

– Figure 2C does not highlight A153, F155? and T255. Is F155 numbering correct? E155 or F155?

2. Related: Amino acid positions for the SH2 domain listed in the Results section are confusing. Authors must indicate what species they are referring to. The figure shows the SH2 domain of vSrc. However, vSrc does not have Glu in either position. It appears that the positions are specified for rat Src. Why did the author choose to take the structure of the vSrc SH2 domain alone? If they would take a look at the structure of full-length human Src (PDB# 1FMK) they would find that both Glu amino acids are positioned in close proximity to the region where the SH3 domain binds suggesting a different mechanism of action if these are the sites spermidine interaction. The docking models shown in Figure 2C, D do not reflect the real interactions because they are done with SH2 alone. They have to be one with full-length Src.

3. Please have a look at the statistical analysis and adjust accordingly:

– Figure 1A. The authors do not indicate how many repeats they did for this experiment. Half of the data points do not have error bars. The error bars for the other half are not specified. The figure legend suggests that the star should indicate a statistically significant difference between samples. However, the samples highlighted in Figure 1A do not appear to be different at all. Also, it is not clear if the other samples are significantly different.

– Figure 1C. The figure legend suggests that the star should indicate a statistically significant difference between samples. However, the samples highlighted in Figure 1C do not appear to be different at all. Also, it is not clear if the other samples are significantly different given how large the error bars are.

– Figure 3C does not show any error bars and does not indicate any statistical analysis.

4. Related: Line 98. Authors stated that "In the presence of fixed amount of spermidine and increasing concentration of the peptide, the maximum rate of Src kinase activity (Vmax) increased, while the affinity (Km) for the substrate was not affected (Figure 1E)." However, neither Vmax nor Km values are provided. Judging from the data, it might be impossible to obtain these parameters within the concentration range tested. Also, the figure legend does not indicate how many replicates were performed and what are the error bars showing. The same criticism applies to Suppl Figure S1.

5. In Figure 1A authors show that wild-type Src is activated two-fold by spermidine. It is important to show how spermidine affects the activity of constitutively active Src. This experiment will also allow authors to evaluate the extent of activation and compare the Kcat and Vmax of constitutively active Src and spermidine-activated. A two-fold increase in activity is not substantial for Src but this could be the limitation of the assay.

6. Figure 1b does not indicate what concentration of spermidine was used in what lane.

7. Authors show that spermidine induced maximum Src activity at 106nM in vitro. However, EC50 in cells is much higher at 6uM. How would the authors explain the discrepancy? Such a drastic difference suggests that the mechanism of Src activation in cells might be different from what they observe with purified protein.

8. The rationale for the experiment in Figure 1D is not clear. Why would there be any activity without ATP or substrate? Some of the bars in Figure 1D are marked with stars indicating statistical significance but it is not clear what they are compared to.

9. Line 129. Authors stated that "Results indicated that Src activity is induced by LPA as measured by the phosphorylation of the Y424, independently of the mutation at the putative allosteric site (Supplementary Figure S3B)." This figure only shows that wild-type Src is activated. Both mutants appear to show elevated activity without LPA and do not show convincing activation following LPA treatment (especially considering uneven Src expression in some samples). The figure legend does not state how many times this experiment was repeated and has no quantification data.

10. Figure 2E shows that spermidine does not activate wt Src contradicting their previous data. Samples without spermidine should be shown on the same blot to compare basal levels of activity. The activity measurements in Figure 2F should compare all samples to the activity of wtSrc without spermidine. This will allow authors to evaluate if mutants have higher residual activity. As mentioned above, mutation of these amino acids may affect the inhibitory binding of the SH3 domain and thus lead to increased basal activity of Src. Same problem with the data in Figure 2H. It shows that the mutants do not respond to spermidine, but it does not reveal if they already have higher activity without spermidine. The figure legend does not provide information about the error bars. Furthermore, the authors suggest that the mechanism of spermidine action is to prevent Src SH2 domain binding phospho-tyrosine. In this case, the luciferase sensor should show a higher signal with higher spermidine concentration regardless of Src activity.

11. Line 159. Authors stated that "Results from immunoblot demonstrated that the co-precipitated IDO1 is tyrosine phosphorylated by Src and that the polyamine increases the phosphorylation (Figure 3A)." The data does not show that Src phosphorylates IDO1. They only suggest that IDO1 phosphorylation increases when Src is co-expressed and activated. Furthermore, there is a band in the IP sample where IDO1 is expressed without Src. This is not explained by the authors. The figure legend says that "IDO1/pTYR ratio is measured by densitometric quantification of the specific bands and is expressed relative to untreated cells." but no measurements are provided.

12. Related: Figure 3, does spermidine promote IDO1 phosphorylation via constitutively active-Src in the SYF model? That spermidine promotes IDO1 phosphorylation is interesting. What advantage/disadvantage does spermidine-mediated IDO1 phosphorylation give to IDO1 function? Does Src co-expression increase IDO1 levels (WCLs, last two lanes Figure 3A) independent of spermidine?

13. The phosphorylation of IDO1 and its interaction with Src upon spermidine treatment are only shown with overexpression of both proteins. To prove a physiological relevance, the effects on endogenous proteins should be evaluated if possible.

14. Is spermidine selective for the IDO1-Src complex? Is it possible additional Src substrates may also be candidates? And are E155 and E174 residues conserved across SH2 domains of other close PTKs, such as Yes and Fyn? Is spermidine binding specifically to the Src SH2 domain? what about SH2 domains of related PTKs? Does spermidine broadly bind/impact SH2 domain-containing protein targets?

15. Is spermidine binding to Src constitutive or induced by stimuli/cues?

*Reviewer #2 (Recommendations for the authors):*

1. Is spermidine selective for the IDO1-Src complex? Is it possible additional Src substrates may also be candidates?

2. Is spermidine binding to Src constitutive or induced by stimuli/cues?

3. Are E155 and E174 residues conserved across SH2 domains of other close PTKs, such as Yes and Fyn? Is spermidine binding specifically to the Src SH2 domain? what about SH2 domains of related PTKs? Does spermidine broadly bind/impact SH2 domain-containing protein targets?

4. It would be more meaningful to include constitutively active-Src and kinase dead-Src in Figure 1A and Figure 1B.

5. Figure S3B, it seems in reconstituted SYF cells, E155A and E174A Src mutants are active in absence of LPA activation and do not significantly respond to LPA ( o vs other time points). A similar profile seems true even with spermidine stimulation of mutants (Figure 2E, F). Why are mutants more active in the LPA experiment (Figure S3B, 0 time points) but not in Figure 2E (-Spd)?

6. Figure 3, does spermidine promote IDO1 phosphorylation via constitutively active-Src in the SYF model? That spermidine promotes IDO1 phosphorylation is interesting. What advantage/disadvantage does spermidine-mediated IDO1 phosphorylation give to IDO1 function? Does Src co-expression increase IDO1 levels (WCLs, last two lanes Figure 3A) independent of spermidine?

*Reviewer #3 (Recommendations for the authors):*

1. Line 65. In the introduction, the authors stated "The autoregulatory function of the kinase occurs through intramolecular interactions that stabilize the catalytically inactive conformation of Src, in which the SH2 domain binds to a pY located at position +535." What Src isoform and species are the authors referring to? cSrc is regulated by phosphorylation of Tyr530 in human, Tyr529 in mouse, Tyr527 in avian protein. Providing a reference would be helpful.

2. Figure 1A. The authors do not indicate how many repeats they did for this experiment. Half of the data points do not have error bars. The error bars for the other half are not specified. The figure legend suggests that the star should indicate a statistically significant difference between samples. However, the samples highlighted in Figure 1A do not appear to be different at all. Also, it is not clear if the other samples are significantly different.

3. In Figure 1A authors show that wild-type Src is activated two-fold by spermidine. It is important to show how spermidine affects the activity of constitutively active Src. This experiment will also allow authors to evaluate the extent of activation and compare the Kcat and Vmax of constitutively active Src and spermidine-activated. A two-fold increase in activity is not substantial for Src but this could be the limitation of the assay.

4. Line 88. The authors stated that they check the phosphorylation of human Src on Tyr424 as an indicator of Src activity. Human Src is phosphorylated on Tyr419, the mouse is on 418, and the avian is on 416.

5. in vitro experiments were done with human Src but there is no clear statement of what Src construct was used in SYF cells.

6. Figure 1b does not indicate what concentration of spermidine was used in what lane.

7. Figure 1C. The figure legend suggests that the star should indicate a statistically significant difference between samples. However, the samples highlighted in Figure 1C do not appear to be different at all. Also, it is not clear if the other samples are significantly different given how large the error bars are.

8. Authors show that spermidine induced maximum Src activity at 106nM in vitro. However, EC50 in cells is much higher at 6uM. How would the authors explain the discrepancy? Such a drastic difference suggests that the mechanism of Src activation in cells might be different from what they observe with purified protein.

9. The rationale for the experiment in Figure 1D is not clear. Why would there be any activity without ATP or substrate? Some of the bars in Figure 1D are marked with stars indicating statistical significance but it is not clear what they are compared to.

10. Line 98. Authors stated that "In the presence of a fixed amount of spermidine and increasing concentration of the peptide, the maximum rate of Src kinase activity (Vmax) increased, while the affinity (Km) for the substrate was not affected (Figure 1E)." However, neither Vmax nor Km values are provided. Judging from the data, it might be impossible to obtain these parameters within the concentration range tested. Also, the figure legend does not indicate how many replicates were performed and what are the error bars showing. The same criticism applies to Suppl Figure S1.

11. Amino acid positions for the SH2 domain listed in the Results section are confusing. Authors must indicate what species they are referring to. The figure shows the SH2 domain of vSrc. However, vSrc does not have Glu in either position. It appears that the positions are specified for rat Src. Why did the author choose to take the structure of the vSrc SH2 domain alone? If they would take a look at the structure of full-length human Src (PDB# 1FMK) they would find that both Glu amino acids are positioned in close proximity to the region where the SH3 domain binds suggesting a different mechanism of action if these are the sites spermidine interaction. The docking models shown in Figure 2C, D do not reflect the real interactions because they are done with SH2 alone. They have to be one with full-length Src.

12. Line 129. Authors stated that "Results indicated that Src activity is induced by LPA as measured by the phosphorylation of the Y424, independently of the mutation at the putative allosteric site (Supplementary Figure S3B)." This figure only shows that wild-type Src is activated. Both mutants appear to show elevated activity without LPA and do not show convincing activation following LPA treatment (especially considering uneven Src expression in some samples). The figure legend does not state how many times this experiment was repeated and has no quantification data.

13. Figure 2E shows that spermidine does not activate wt Src contradicting their previous data. Samples without spermidine should be shown on the same blot to compare basal levels of activity. The activity measurements in Figure 2F should compare all samples to the activity of wtSrc without spermidine. This will allow authors to evaluate if mutants have higher residual activity. As mentioned above, mutation of these amino acids may affect the inhibitory binding of the SH3 domain and thus lead to increased basal activity of Src. Same problem with the data in Figure 2H. It shows that the mutants do not respond to spermidine, but it does not reveal if they already have higher activity without spermidine. The figure legend does not provide information about the error bars. Furthermore, the authors suggest that the mechanism of spermidine action is to prevent Src SH2 domain binding phospho-tyrosine. In this case, the luciferase sensor should show a higher signal with higher spermidine concentration regardless of Src activity.

14. Line 159. Authors stated that "Results from immunoblot demonstrated that the co-precipitated IDO1 is tyrosine phosphorylated by Src and that the polyamine increases the phosphorylation (Figure 3A)." The data does not show that Src phosphorylates IDO1. They only suggest that IDO1 phosphorylation increases when Src is co-expressed and activated. Furthermore, there is a band in the IP sample where IDO1 is expressed without Src. This is not explained by the authors. The figure legend says that "IDO1/pTYR ratio is measured by densitometric quantification of the specific bands and is expressed relative to untreated cells." but no measurements are provided.

15. Figure 3C does not show any error bars and does not indicate any statistical analysis.

16. The phosphorylation of IDO1 and its interaction with Src upon spermidine treatment are only shown with overexpression of both proteins. To prove a physiological relevance, the effects on endogenous proteins should be evaluated.

---

## [Author Response]

Essential revisions:1. Please be more clear in the numbering of amino acids and specific species used:1a. Line 65. In the introduction, the authors stated "The autoregulatory function of the kinase occurs through intramolecular interactions that stabilize the catalytically inactive conformation of Src, in which the SH2 domain binds to a pY located at position +535." What Src isoform and species are the authors referring to? cSrc is regulated by phosphorylation of Tyr530 in human, Tyr529 in mouse, Tyr527 in avian protein. Providing a reference would be helpful.

The Authors wish to sincerely apologize for the presence of incorrect statements. Numbering of amino acids has been revised thorough the manuscript, indicating residues of the human Src kinase isoform. When other isoforms (viral and/or murine) are cited, numbering of the residues according to the relative sequence is also explicitly indicated. At line 67, the text has been now corrected accordingly, indicating Tyr530 of the human sequence.

1b. Line 88. The authors stated that they check the phosphorylation of human Src on Tyr424 as an indicator of Src activity. Human Src is phosphorylated on Tyr419, the mouse is on 418, and avian is on 416.

We apologize for the mistake and now the text (line 89) has been modified accordingly, indicating Tyr418 of the murine sequence.

1c. in vitro experiments were done with human Src but there is no clear statement of what Src construct was used in SYF cells.

In response to the referee’s request, we specify that the experiments with SYF cells were performed by using the murine Src construct, while biochemical assays were done with recombinant human Src protein. The text has been modified accordingly, in both results and material/methods sections.

1d. Figure 2C does not highlight A153, F155? and T255. Is F155 numbering correct? E155 or F155?

The Authors wish to sincerely apologize for the presence of such mistakes. In response to Reviewer request, Figures 2C and 2D and their legends have been modified. Highlighted residues are now labelled according to sequence numbering of the human isoform. In the legend of Figure 2B, the corresponding residues of the NMR structure of viral SH2 domain are also indicated.

2. Related: Amino acid positions for the SH2 domain listed in the Results section are confusing. Authors must indicate what species they are referring to. The figure shows the SH2 domain of vSrc. However, vSrc does not have Glu in either position. It appears that the positions are specified for rat Src.

The Authors wish to sincerely apologize for the presence of incorrect numbering. Position of residues of the SH2 domain listed in the Results section has been revised according to sequence numbering of the human isoform. When the murine isoform is cited (results of mutagenesis experiments), numbering of the residues according to the relative sequence is also explicitly indicated.

2a. Why did the author choose to take the structure of the vSrc SH2 domain alone? If they would take a look at the structure of full-length human Src (PDB# 1FMK) they would find that both Glu amino acids are positioned in close proximity to the region where the SH3 domain binds suggesting a different mechanism of action if these are the sites spermidine interaction. The docking models shown in Figure 2C, D do not reflect the real interactions because they are done with SH2 alone. They have to be one with full-length Src.

We understand the concern of the referee. We used the NMR structure of the SH2 domain of Src kinase of Rous sarcoma virus (PDB ID: 2JYQ) because it shares a high sequence identity (97%) with the relative SH2 domains of human and murine Src tyrosine kinase. This is now explicitly stated in the Materials and methods section of the revised manuscript, also inserting the sequence of the viral isoform in the multiple alignment reported in Figure 2 —figure supplement 1. The very high sequence identity allows using the NMR structure of the viral isoform with very good approximation to infer about the electrostatic potential map of human/murine SH2 domain and to investigate the putative binding site of spermidine.

The advantage of using a single domain structure rather than the full-length structure of human Src (PDB# 1FMK) is to avoid the bias that a full-length structure of human Src kinase might bring into more time-consuming calculations due to considering multiple conformations between the regulatory and kinase domains of the enzyme. Indeed, diverse conformations of Src may differently affect the electrostatic potential of the SH2 domain as well as potentially mask a putative polyamine binding site for polyamine.

Although we agree with Reviewer’s comment that the human full-length Src may be instrumental to provide more insights into the mechanism of action of spermidine, this latter was not the aim of the present study. Indeed, using electrostatic potential mapping and docking calculations, the goal of the study was to generate a working hypothesis to locate a potential binding site of spermidine in the SH2 domain and next prove such hypothesis with mutagenesis experiments. Results of mutagenesis experiments are in agreement with the working hypothesis generated using the NMR structure of the viral SH2 domain, thereby supporting the validity of using such isoform as input structure for our calculations.

3. Please have a look at the statistical analysis and adjust accordingly:3a. Figure 1A. The authors do not indicate how many repeats they did for this experiment. Half of the data points do not have error bars. The error bars for the other half are not specified. The figure legend suggests that the star should indicate a statistically significant difference between samples. However, the samples highlighted in Figure 1A do not appear to be different at all. Also, it is not clear if the other samples are significantly different.

We understand the concern of the Referee about the error bars. In this figure, no error bar appears for certain points because they are shorter than the size of the symbol. Thus, to overcome this limitation, we modified the Figure 1A by making the symbols smaller. In the figure legend we specified that the results are the mean of three independent experiments, each performed in triplicates. Moreover, we indicated that the statistical analysis was performed by comparing the kinase activity of each spermidine-treated sample to the untreated counterpart.

3b. Figure 1C. The figure legend suggests that the star should indicate a statistically significant difference between samples. However, the samples highlighted in Figure 1C do not appear to be different at all. Also, it is not clear if the other samples are significantly different given how large the error bars are.

In response to the Referee request, we specify that the data were analyzed by comparing the pSrc/Src ratio of each spermidine-treated sample to the untreated counterpart (the legend of Figure 1C has been modified accordingly). Although the potentiating effect of spermidine on Src activity in SYF cells is relatively subtle, it is nonetheless likely to be physiologically relevant, as small degrees of positive allosteric modulation of the Src by spermidine is known to be relevant in conventional dendritic cells (Mondanelli ed al., Immunity, 2017).

3c. Figure 3C does not show any error bars and does not indicate any statistical analysis.

We wish to apologize for this oversight. As suggested by the Referee, Revised Figure 3C now shows the error bars indicating the standard deviation and the statistical analysis, which has been performed by comparing, for each time point, the pTYR/IDO1 ratio of spermidine-treated sample to the untreated counterpart (as specified in the figure legend).

4. Related: Line 98. Authors stated that "In the presence of fixed amount of spermidine and increasing concentration of the peptide, the maximum rate of Src kinase activity (Vmax) increased, while the affinity (Km) for the substrate was not affected (Figure 1E)." However, neither Vmax nor Km values are provided. Judging from the data, it might be impossible to obtain these parameters within the concentration range tested. Also, the figure legend does not indicate how many replicates were performed and what are the error bars showing. The same criticism applies to Suppl Figure S1.

We agree with the point raised by the Referee, i.e., the limited concentration range tested for ATP and peptide. We thus tried to use higher concentrations of either ATP or peptide. The revised Figure 1F and Revised Figure 1 —figure supplement 1 now show the curve of peptide or ATP from 0 to 600 µM. Due to the limit of detection of the luminescent assay as well as to the saturating effect, we were not able to include any other concentrations higher than 600 µM. The results confirmed that in the presence of increasing concentrations of peptide (but not in the case of increasing concentrations of ATP), spermidine enhanced by almost 2-fold the maximum rate of Src kinase activity and the affinity for the substrate. Moreover, as indicated by the Referee, we specified the Vmax and Km values in the revised figures as well as the number of experiments performed.

5. In Figure 1A authors show that wild-type Src is activated two-fold by spermidine. It is important to show how spermidine affects the activity of constitutively active Src. This experiment will also allow authors to evaluate the extent of activation and compare the Kcat and Vmax of constitutively active Src and spermidine-activated. A two-fold increase in activity is not substantial for Src but this could be the limitation of the assay.

We understand the point raised by the Reviewer and his/her skepticism on the polyamine effect. We got the importance of showing that spermidine increases the activity of the constitutively active Src. However, due to funding shortage and long time required for the reagent purchase and shipment, we were not able to perform the specific experiment with the recombinant viral Src protein as required by the Referee. We wish to sincerely apologize for that.

Nevertheless, we performed related experiments with SYF cells expressing the wild type Src or the mutated one at the tyrosine 529 into phenylalanine (Y529F of the murine sequence) that is constitutive active Src. By measuring the pSrc/Src ratio (revised Figure 1 —figure supplement 1) as well as the phosphorylation of IDO1 (revised Figure 3 —figure supplement 1) in cells treated with spermidine, we found that spermidine did not potentiate the activity of Src Y529F.

We thus concluded that, upon binding the allosteric site on the SH2 domain, spermidine makes Src to change its tridimensional structure. We speculated that spermidine promotes the conformational changes of the kinase and thus its activation. The results obtained with the SYF model highlight that spermidine per se cannot activate or stabilize the constitutive-active form of Src, i.e., that mutated in the residue Y529 at the C-terminus. Therefore, spermidine behaves as a pharmacologic allosteric modulator of Src, as it binds to a distinct site from the catalytic pocket and guides Src to assume the active conformation.

We understand the concern about the 2-fold increase in activity. We would like to stress that this result has been obtained by means of biochemical and fibroblast-based assays, because the aim of this paper was to show the spermidine mechanism of action with these two in vitro models. However, a 2-fold increment of Src activity is sufficient to phosphorylate IDO1 in dendritic cells and finally promote the immunoregulatory phenotype of such immune cells as demonstrated in Mondanelli et al. (Immunity, 2017) and in line with the concept of allosteric modulation.

6. Figure 1b does not indicate what concentration of spermidine was used in what lane.

In response to Referee request, we modified the Figure 1B accordingly, by indicating the concentration of spermidine used.

7. Authors show that spermidine induced maximum Src activity at 106nM in vitro. However, EC50 in cells is much higher at 6uM. How would the authors explain the discrepancy? Such a drastic difference suggests that the mechanism of Src activation in cells might be different from what they observe with purified protein.

We understand the concern of the referee. However, given the intrinsic difference between the cell-free (i.e., the biochemical assay with recombinant Src protein) and cell-based assay (i.e., that with SYF fibroblasts reconstituted with murine Src vector), the tested compound can exhibit distinct potency. Indeed, in a cell-based assay the molecule must penetrate through the cell membrane, distribute within the cell, find the target among the other proteins, and finally activate it. On the contrary, in a biochemical assay, the molecule is directly added to the reaction mixture containing only the purified target protein (besides other reagents in the optimal concentration to carry out the in vitro reaction). Thus, is not surprising that the EC50 of spermidine in the SYF cellbased assay is higher than that found in the biochemical assay.

8. The rationale for the experiment in Figure 1D is not clear. Why would there be any activity without ATP or substrate? Some of the bars in Figure 1D are marked with stars indicating statistical significance but it is not clear what they are compared to.

The point raised by the Reviewer is well-taken. The experiments without ATP or peptide are negative controls of the biochemical assay. Our aim was to confirm the lack of intrinsic activity of the molecule spermidine against purified Src when neither the ATP nor the substrate are present, thus measuring the background signal – which is below 1000 rlu. On the contrary, the activity of Src in the samples with ATP and peptide is above 10000 rlu, which at least doubles in the presence of spermidine. In accordance with the Reviewer #1 request (please, see Reviewer #1, point 3), the statistical analysis of data in Figure 1D (now revised Figure 1E) has been now performed using the Student t-test, by comparing Spd/ATP/ peptide vs ATP/peptide samples. The revised analysis still confirms that the presence of spermidine increases the kinase activity by 2-fold. The Figure and the respective legend have been modified accordingly.

9. Line 129. Authors stated that "Results indicated that Src activity is induced by LPA as measured by the phosphorylation of the Y424, independently of the mutation at the putative allosteric site (Supplementary Figure S3B)." This figure only shows that wild-type Src is activated. Both mutants appear to show elevated activity without LPA and do not show convincing activation following LPA treatment (especially considering uneven Src expression in some samples). The figure legend does not state how many times this experiment was repeated and has no quantification data.

We apologize for the bad quality of the Western blot experiment shown in Supplementary Figure S3. We have now repeated the experiment and the results are now shown in revised Figure 2 —figure supplement 2, alongside the densitometric analysis demonstrating that LPA activates Src in cells expressing the mutant forms of the kinase similarly to the wild-type protein (revised Figure 2 —figure supplement 2).

Moreover, as shown by the revised Figure 2 —figure supplement 2, the mutant forms of the kinase have a basal activity comparable to wild-type protein. The same results can be obtained by looking at the absolute values of luminescent reporter in untreated SYF-Src cells as well as those expressing Src E149A or Src E168A (please, see lines 149-153 of the main text). Specifically, luminescent signal in unstimulated SYF cells co-expressing the reporter and wild type Src is 617,7 ± 121 rlu vs 552,2 ± 68,9 rlu of E149A Src vs 501 ± 110 rlu of E168A Src – which are not statistically different (one-way ANOVA, followed by post-hoc Bonferroni test).

10. Figure 2E shows that spermidine does not activate wt Src contradicting their previous data. Samples without spermidine should be shown on the same blot to compare basal levels of activity. The activity measurements in Figure 2F should compare all samples to the activity of wtSrc without spermidine. This will allow authors to evaluate if mutants have higher residual activity. As mentioned above, mutation of these amino acids may affect the inhibitory binding of the SH3 domain and thus lead to increased basal activity of Src.

As suggested by the Reviewer, we evaluated the activity of Src in unstimulated cells expressing the wild-type protein or the mutated one. As shown in the revised Figure 2 —figure supplement 2, the amino acid substitution at the position 149 and 168 does not affect the basal activity of the kinase, thus suggesting that the Src mutants can be used to validate the working hypothesis generated through the electrostatic potential mapping and docking calculations. Moreover, to overcome the limitation of the samples loaded on different blots, we have now repeated the experiments running samples stimulated with selected spermidine concentration (i.e., 100 µM) in the same gel, so as to have all the cell types. The results are now shown in revised Figure 2E.

10a. Same problem with the data in Figure 2H. It shows that the mutants do not respond to spermidine, but it does not reveal if they already have higher activity without spermidine.

As reported in point 9 of the “Essential revision for authors”, the absolute values of luminescent reporter in untreated cells co-expressing either the wild type Src or those expressing Src E149A or Src E168A are not statistically different, suggesting that the basal activity of the kinase is not affected by the specific amino acid substitution. Specifically, luminescent signal in unstimulated SYF cells co-expressing the reporter and wild type Src is 617,7 ± 121 rlu vs 552,2 ± 68,9 rlu of E149A Src vs 501 ± 110 rlu of E168A Src – which are not statistically different (one-way ANOVA, followed by post-hoc Bonferroni test). This information has been specified in the main text (please see lines 149 – 153).

10b. The figure legend does not provide information about the error bars.

As suggested by the Referee, the figure legend has been modified indicating that the results in Figure 2H are mean ± standard deviation of three independent experiments.

10c. Furthermore, the authors suggest that the mechanism of spermidine action is to prevent Src SH2 domain binding phospho-tyrosine. In this case, the luciferase sensor should show a higher signal with higher spermidine concentration regardless of Src activity.

The bioluminescent reporter has been used as an alternative strategy to demonstrate the capability of spermidine to activate the wild type Src kinase expressed by SYF cells, and not the mutated one. Indeed, the DNA plasmid-based reporter is built by physically separated the N and C luciferase fragments introducing a phosphopeptide recognition domain (i.e., the SH2 sequence) and the Src peptide substrate. Upon cellular Src activation, the reporter peptide substrate becomes phosphorylated, interacts with SH2 of the reporter, thus creating a steric hindrance that prevents luciferase reconstitution and bioluminescence emission (please, refer to the schematic representation Figure 2G). Thus, in the presence of spermidine, the luciferase sensor shows a lower signal in cells co-expressing the wild-type Src and not the mutated forms of the kinase, suggesting that the polyamine activates only the cellular, wild-type Src by acting at the putative allosteric site.

11. Line 159. Authors stated that "Results from immunoblot demonstrated that the co-precipitated IDO1 is tyrosine phosphorylated by Src and that the polyamine increases the phosphorylation (Figure 3A)." The data does not show that Src phosphorylates IDO1. They only suggest that IDO1 phosphorylation increases when Src is co-expressed and activated.

We agree with the Reviewer that Figure 3A shows that IDO1 phosphorylation increases when Src is co-expressed and activated by spermidine, thus we have modified the main text accordingly (please see line 176-177).

11a. Furthermore, there is a band in the IP sample where IDO1 is expressed without Src. This is not explained by the authors.

The band in the IP sample where IDO1 is expressed without Src has molecular weight of around 40 kDa, lower than that of pIDO1. We might speculate that is a non-specific signal. In response to Referee request (please see the minor point 10), we have now replaced the figure with a revised version of IP experiment that includes the negative control (i.e., the immunoprecipitation without antibody; revised Figure 3A) as well.

11b. The figure legend says that "IDO1/pTYR ratio is measured by densitometric quantification of the specific bands and is expressed relative to untreated cells." but no measurements are provided.

As stated in the figure legend, panel A also included the densitometric quantification of the amount of IDO1 co-precipitated in samples after immunoprecipitation with the a-pTYR antibody. The measurement is reported under the specific IP bands and is expressed relative to untreated cells (fold change = 1). To be clearer, we modified the statement in the figure legend as follows: “The amount of IDO1 immunoprecipitated is measured by densitometric quantification of the specific bands in treated sample co-expressing IDO1 and Src and is expressed relative to untreated cells (fold change = 1)”, as the analysis refers to the amount of IDO1 precipitated with the a-pTyr antibody.

12. Related: Figure 3, does spermidine promote IDO1 phosphorylation via constitutively active-Src in the SYF model?

We would like to thank the Reviewer for this suggestion. By immunoprecipitation with the a-pTyr antibody, we found that IDO1 is phosphorylated in SYF cells co-expressing IDO1 and the constitutively active-Src regardless of the stimulation with spermidine, as opposed to samples coexpressing IDO1 and wild type Src (please, refer to revised Figure 3 —figure supplement 1). This result suggests that spermidine can act as on/off switcher. Indeed, when the polyamine binds the site on the SH2 domain, which is distinct from the substrate pocket, it can make Src assume an active conformation, essentially turning it on. Conversely, constitutive active Src adopts already the open conformation, as the substitution of the tyrosine residue at position 529 of the murine sequence with the phenylalanine interferes with the autoregulatory clamp of the enzyme. In that case, spermidine does not increase the Src activity – as also shown by the measurement of the pSrc/Src in SYF cells exposed to spermidine and expressing constitutively active Src (Revised Figure 1 —figure supplement 1).

That spermidine promotes IDO1 phosphorylation is interesting. What advantage/disadvantage does spermidine-mediated IDO1 phosphorylation give to IDO1 function?

We would like to thank the Reviewer for highlighting this point. We have previously demonstrated that spermidine confers an immunoregulatory phenotype on conventional dendritic cells (i.e., the antigen presenting cells of our immune system; cDCs) via IDO1 (Mondanelli et al., Immunity, 2017). Specifically, we have shown that spermidine promotes the phosphorylation of IDO1 and the activation of its non-enzymatic function in cDCs. However, the exact molecular mechanism behind it all was the missing puzzle piece. Through the present study, we thus wondered whether Src directly phosphorylates IDO1 and if spermidine promotes such post-translational modification by resorting to SYF cells appropriately reconstituted with the vectors coding for Src and IDO1 as in vitro model.

Does Src co-expression increase IDO1 levels (WCLs, last two lanes Figure 3A) independent of spermidine?

We understand the concern raised by the Reviewer. The experiment represented in Figure 3A has been performed on SYF cells reconstituted with vectors coding for Src and IDO1, thus we may exclude that the co-expression of Src could increase IDO1 levels.

13. The phosphorylation of IDO1 and its interaction with Src upon spermidine treatment are only shown with overexpression of both proteins. To prove a physiological relevance, the effects on endogenous proteins should be evaluated if possible.

We would like to thank the Reviewer for this suggestion. We have now performed the experiments on colon cancer cell line that express both IDO1 and Src kinase (i.e., MC38). Results showed that spermidine activates Src kinase as measured by its phosphorylation status (revised Figure 1D). Moreover, spermidine promotes the phosphorylation of IDO1 and its interaction with Src (revised Figure 3H, 3I).

14. Is spermidine selective for the IDO1-Src complex? Is it possible additional Src substrates may also be candidates?

We would like to thank the Reviewer for highlighting this point. Among the predicted functional partners of murine Src (https://string-db.org/), we verified Stat3 as possible candidate. Results demonstrated that spermidine does not increase the interaction of Src with Stat3 in SYF cells, as depicted in Author response image 1. Although we cannot exclude the effect on any other candidates, our analysis indicated that spermidine is selective for the IDO1-Src complex. These findings are in line with the notion that allosteric ligands can selectively affect only one biological process and have no effect on any other response via the pharmacologic target. These results have been included as “data not shown” in the main text (please see lines 189-191).

**Author response image 1. sa2fig1:** 

14a. And are E155 and E174 residues conserved across SH2 domains of other close PTKs, such as Yes and Fyn? Is spermidine binding specifically to the Src SH2 domain? what about SH2 domains of related PTKs? Does spermidine broadly bind/impact SH2 domain-containing protein targets?

The point raised by the Reviewer is well taken. The alignment of murine Src, Yes and Fyn sequences highlights that only the glutamate residue at position 149 of murine Src is conserved across SH2 domains, while the glutamate residue at position 168 is replaced by the amino acid glycine, as depicted in Author response image 2. Thus, although we cannot exclude that spermidine can bind the SH2 domain of related Src kinases, we might speculate that the lack of the negative charge at position 168 reduces the long-range electrostatic interactions hypothesized to be responsible for the molecular recognition of the ligand into the allosteric site. These findings were included in the main text as data not shown (please see lines 162-167).

By aligning murine Src with several SH2 domain proteins such as enzymes (Abl1, SHIP1), docking (Shc1) and adaptor (Grb2, SOCS1) proteins, we found that neither E149 nor E168 are conserved, in their place there are non-polar hydrophobic amino acids (e.g., alanine, glycine and proline) or those with polar uncharged side chains (e.g., glutamine). In light of these results, we may conclude that spermidine cannot broadly bind SH2-domain containing proteins, suggesting that the polyamine selectively interacts with Src kinase. These observations have been included in the main text (please see lines 250-255). Please, refer to Author response image 2 for sequences alignment by CLUSTAL.

15. Is spermidine binding to Src constitutive or induced by stimuli/cues?

As shown in the biochemical assay with the recombinant protein (please see Figure 1A) as well as in SYF cells reconstituted with the construct (please see Figure 1B-C), we conclude that spermidine binding to Src does not depend on other general cues.

Reviewer #2 (Recommendations for the authors):1. Is spermidine selective for the IDO1-Src complex? Is it possible additional Src substrates may also be candidates?

See Essential revisions response to point 14.

2. Is spermidine binding to Src constitutive or induced by stimuli/cues?

See Essential revisions response to point 15.

3. Are E155 and E174 residues conserved across SH2 domains of other close PTKs, such as Yes and Fyn?

See Essential revisions response to point 14a.

Is spermidine binding specifically to the Src SH2 domain? what about SH2 domains of related PTKs? Does spermidine broadly bind/impact SH2 domain-containing protein targets?

By aligning murine Src with several SH2 domain proteins such as enzymes (Abl1, SHIP1), docking (Shc1) and adaptor (Grb2, SOCS1) proteins, we found that neither E149 nor E168 are conserved, in their place there are non-polar hydrophobic amino acids (e.g., alanine, glycine and proline) or those with polar uncharged side chains (e.g., glutamine). In light of these results, we may conclude that spermidine cannot broadly bind SH2-domain containing proteins, suggesting that the polyamine selectively interacts with Src kinase. These observations have been included in the main text (please see lines 250-255). Please, refer to attached files of sequences alignment by CLUSTAL.

4. It would be more meaningful to include constitutively active-Src and kinase dead-Src in Figure 1A and Figure 1B.

We understand the point raised by the Reviewer. However, due to funding shortage and long time required for the reagent purchase and shipment, we were not able to perform the specific experiment with the recombinant proteins as required by the Referee (as parallel experiments for figure 1A). We wish to sincerely apologize for that.

Nevertheless, we performed related experiments with SYF cells expressing the wild type Src or the mutated one at the tyrosine 529 into phenylalanine (Y529F of the murine sequence) that is constitutively active Src. By measuring the pSrc/Src ratio (Figure 1 —figure supplement 1) as well as the phosphorylation of IDO1 (Figure 3 —figure supplement 1) in cells treated with spermidine, we found that spermidine did not potentiate the activity of Src Y529F.

We thus concluded that, upon binding the allosteric site on the SH2 domain, spermidine makes Src to change its tridimensional structure. We speculated that spermidine promotes the conformational changes of the kinase and thus its activation. The results obtained with the SYF model highlight that spermidine per se cannot activate or stabilize the constitutive-active form of Src, i.e., that mutated in the residue Y529 at the C-terminus. Therefore, spermidine behaves as a pharmacologic allosteric modulator of Src, as it binds to a distinct site from the catalytic pocket and guides Src to assume the active conformation.

5. Figure S3B, it seems in reconstituted SYF cells, E155A and E174A Src mutants are active in absence of LPA activation and do not significantly respond to LPA ( o vs other time points). A similar profile seems true even with spermidine stimulation of mutants (Figure 2E, F). Why are mutants more active in the LPA experiment (Figure S3B, 0 time points) but not in Figure 2E (-Spd)?

We apologize for the bad quality of the Western blot experiment shown in Supplementary Figure S3. We have now repeated the experiment and the results are now shown in revised Figure 2 —figure supplement 2, alongside the densitometric analysis demonstrating that LPA activates Src in cells expressing the mutant forms of the kinase similarly to the wild-type protein (Figure 2 —figure supplement 2).

Moreover, as shown by the revised Figure 2 —figure supplement 2, the mutant forms of the kinase have a basal activity comparable to wild-type protein. The same results can be obtained by looking at the absolute values of luminescent reporter in untreated SYF-Src cells as well as those expressing Src E149A or Src E168A (please, see lines 149-153 of the main text). Specifically, luminescent signal in unstimulated SYF cells co-expressing the reporter and wild type Src is 617,7 ± 121 rlu vs 552,2 ± 68,9 rlu of E149A Src vs 501 ± 110 rlu of E168A Src – which are not statistically different (one-way ANOVA, followed by post-hoc Bonferroni test).

6. Figure 3, does spermidine promote IDO1 phosphorylation via constitutively active-Src in the SYF model?

We would like to thank the Reviewer for this suggestion. By immunoprecipitation with the a-pTyr antibody, we found that IDO1 is phosphorylated in SYF cells co-expressing IDO1 and the constitutively active-Src regardless of the stimulation with spermidine, as opposed to samples coexpressing IDO1 and wild type Src (please, refer to revised Figure 3 —figure supplement 1). This result suggests that spermidine can act as on/off switcher. Indeed, when the polyamine binds the site on the SH2 domain, which is distinct from the substrate pocket, it can make Src assume an active conformation, essentially turning it on. Conversely, constitutive active Src adopts already the open conformation, as the substitution of the tyrosine residue at position 529 of the murine sequence with the phenylalanine interferes with the autoregulatory clamp of the enzyme. In that case, spermidine does not increase the Src activity – as also shown by the measurement of the pSrc/Src in SYF cells exposed to spermidine and expressing constitutively active Src (Revised Figure 1 —figure supplement 1).

That spermidine promotes IDO1 phosphorylation is interesting. What advantage/disadvantage does spermidine-mediated IDO1 phosphorylation give to IDO1 function?

We would like to thank the Reviewer for highlighting this point. We have previously demonstrated that spermidine confers an immunoregulatory phenotype on conventional dendritic cells (i.e., the antigen presenting cells of our immune system; cDCs) via IDO1 (Mondanelli et al., Immunity, 2017). Specifically, we have shown that spermidine promotes the phosphorylation of IDO1 and the activation of its non-enzymatic function in cDCs. However, the exact molecular mechanism behind it all was the missing puzzle piece. Through the present study, we thus wondered whether Src directly phosphorylates IDO1 and if spermidine promotes such post-translational modification by resorting to SYF cells appropriately reconstituted with the vectors coding for Src and IDO1 as an in vitro model.

Does Src co-expression increase IDO1 levels (WCLs, last two lanes Figure 3A) independent of spermidine?

We understand the concern raised by the Reviewer. The experiment represented in Figure 3A has been performed on SYF cells reconstituted with vectors coding for Src and IDO1, thus we may exclude that the co-expression of Src could increase IDO1 levels.

Reviewer #3 (Recommendations for the authors):1. Line 65. In the introduction, the authors stated "The autoregulatory function of the kinase occurs through intramolecular interactions that stabilize the catalytically inactive conformation of Src, in which the SH2 domain binds to a pY located at position +535." What Src isoform and species are the authors referring to? cSrc is regulated by phosphorylation of Tyr530 in human, Tyr529 in mouse, Tyr527 in avian protein. Providing a reference would be helpful.

See response to Essential revisions point 1a.

2. Figure 1A. The authors do not indicate how many repeats they did for this experiment. Half of the data points do not have error bars. The error bars for the other half are not specified. The figure legend suggests that the star should indicate a statistically significant difference between samples. However, the samples highlighted in Figure 1A do not appear to be different at all. Also, it is not clear if the other samples are significantly different.

See response to Essential revisions point 3a.

3. In Figure 1A authors show that wild-type Src is activated two-fold by spermidine. It is important to show how spermidine affects the activity of constitutively active Src. This experiment will also allow authors to evaluate the extent of activation and compare the Kcat and Vmax of constitutively active Src and spermidine-activated. A two-fold increase in activity is not substantial for Src but this could be the limitation of the assay.

See response to Essential revisions point 5.

4. Line 88. The authors stated that they check the phosphorylation of human Src on Tyr424 as an indicator of Src activity. Human Src is phosphorylated on Tyr419, the mouse is on 418, and the avian is on 416.

See response to Essential revisions point 1b.

5. in vitro experiments were done with human Src but there is no clear statement of what Src construct was used in SYF cells.

In response to the referee’s request, we specify that the experiments with SYF cells were performed by using the murine Src construct, while biochemical assays were done with recombinant human Src protein. The text has been modified accordingly, in both results and material & methods sections.

6. Figure 1b does not indicate what concentration of spermidine was used in what lane.

In response to Referee request, we modified the Figure 1B accordingly by indicating the concentration of spermidine used.

7. Figure 1C. The figure legend suggests that the star should indicate a statistically significant difference between samples. However, the samples highlighted in Figure 1C do not appear to be different at all. Also, it is not clear if the other samples are significantly different given how large the error bars are.

See response to Essential revisions point 3b.

8. Authors show that spermidine induced maximum Src activity at 106nM in vitro. However, EC50 in cells is much higher at 6uM. How would the authors explain the discrepancy? Such a drastic difference suggests that the mechanism of Src activation in cells might be different from what they observe with purified protein.

See response to Essential revisions point 7.

9. The rationale for the experiment in Figure 1D is not clear. Why would there be any activity without ATP or substrate? Some of the bars in Figure 1D are marked with stars indicating statistical significance but it is not clear what they are compared to.

See response to Essential revisions point 8.

10. Line 98. Authors stated that "In the presence of a fixed amount of spermidine and increasing concentration of the peptide, the maximum rate of Src kinase activity (Vmax) increased, while the affinity (Km) for the substrate was not affected (Figure 1E)." However, neither Vmax nor Km values are provided. Judging from the data, it might be impossible to obtain these parameters within the concentration range tested. Also, the figure legend does not indicate how many replicates were performed and what are the error bars showing. The same criticism applies to Suppl Figure S1.

See response to Essential revisions point 4.

11. Amino acid positions for the SH2 domain listed in the Results section are confusing. Authors must indicate what species they are referring to. The figure shows the SH2 domain of vSrc. However, vSrc does not have Glu in either position. It appears that the positions are specified for rat Src.

See response to Essential revisions point 2.

Why did the author choose to take the structure of the vSrc SH2 domain alone? If they would take a look at the structure of full-length human Src (PDB# 1FMK) they would find that both Glu amino acids are positioned in close proximity to the region where the SH3 domain binds suggesting a different mechanism of action if these are the sites spermidine interaction. The docking models shown in Figure 2C, D do not reflect the real interactions because they are done with SH2 alone. They have to be one with full-length Src.

See response to Essential revisions point 2a.

12. Line 129. Authors stated that "Results indicated that Src activity is induced by LPA as measured by the phosphorylation of the Y424, independently of the mutation at the putative allosteric site (Supplementary Figure S3B)." This figure only shows that wild-type Src is activated. Both mutants appear to show elevated activity without LPA and do not show convincing activation following LPA treatment (especially considering uneven Src expression in some samples). The figure legend does not state how many times this experiment was repeated and has no quantification data.

See response to Essential revisions point 9.

13. Figure 2E shows that spermidine does not activate wt Src contradicting their previous data. Samples without spermidine should be shown on the same blot to compare basal levels of activity. The activity measurements in Figure 2F should compare all samples to the activity of wtSrc without spermidine. This will allow authors to evaluate if mutants have higher residual activity. As mentioned above, mutation of these amino acids may affect the inhibitory binding of the SH3 domain and thus lead to increased basal activity of Src.

See response to Essential revisions point 10.

Same problem with the data in Figure 2H. It shows that the mutants do not respond to spermidine, but it does not reveal if they already have higher activity without spermidine.

See response to Essential revisions point 10a.

The figure legend does not provide information about the error bars.

See response to Essential revisions point 10b.

Furthermore, the authors suggest that the mechanism of spermidine action is to prevent Src SH2 domain binding phospho-tyrosine. In this case, the luciferase sensor should show a higher signal with higher spermidine concentration regardless of Src activity.

See response to Essential revisions point 10c.

14. Line 159. Authors stated that "Results from immunoblot demonstrated that the co-precipitated IDO1 is tyrosine phosphorylated by Src and that the polyamine increases the phosphorylation (Figure 3A)." The data does not show that Src phosphorylates IDO1. They only suggest that IDO1 phosphorylation increases when Src is co-expressed and activated.

See response to Essential revisions point 11.

Furthermore, there is a band in the IP sample where IDO1 is expressed without Src. This is not explained by the authors.

See response to Essential revisions point 11a.

The figure legend says that "IDO1/pTYR ratio is measured by densitometric quantification of the specific bands and is expressed relative to untreated cells." but no measurements are provided.

See response to Essential revisions point 11b.

15. Figure 3C does not show any error bars and does not indicate any statistical analysis.

See response to Essential revisions point 3c.

16. The phosphorylation of IDO1 and its interaction with Src upon spermidine treatment are only shown with overexpression of both proteins. To prove a physiological relevance, the effects on endogenous proteins should be evaluated.

See response to Essential revisions point 13.